# Genome-wide quantification of the effects of DNA methylation on human gene regulation

Amanda J Lea[1†*], Christopher M Vockley[2,3‡], Rachel A Johnston[4], Christina A Del Carpio[4§], Luis B Barreiro[5#], Timothy E Reddy[2,3,6], Jenny Tung[1,4,7,8*]

[1]Department of Biology, Duke University, North Carolina, United States; [2]Center for Genomic and Computational Biology, Duke University Medical School, North Carolina, United States; [3]Department of Biostatistics and Bioinformatics, Duke University Medical School, North Carolina, United States; [4]Department of Evolutionary Anthropology, Duke University, North Carolina, United States; [5]Department of Pediatrics, Sainte-Justine Hospital Research Centre, University of Montreal, Montreal, Canada; [6]Program in Computational Biology and Bioinformatics, Duke University, North Carolina, United States; [7]Institute of Primate Research, National Museums of Kenya, Nairobi, Kenya; [8]Duke University Population Research Institute, Duke University, North Carolina, United States

*For correspondence:
alea@princeton.edu (AJL);
jt5@duke.edu (JT)

Present address: [†]Lewis-Sigler Institute for Integrative Genomics, Carl Icahn Laboratory, Princeton, United States; [‡]Broad Institute of MIT and Harvard, Cambridge, United States; [§]Department of Ecology and Evolutionary Biology, University of California, Los Angeles, United States; [#]Department of Medicine, Section of Genetic Medicine, University of Chicago, Chicago, United States

Competing interests: The authors declare that no competing interests exist.

**Abstract** Changes in DNA methylation are involved in development, disease, and the response to environmental conditions. However, not all regulatory elements are functionally methylation-dependent (MD). Here, we report a method, mSTARR-seq, that assesses the causal effects of DNA methylation on regulatory activity at hundreds of thousands of fragments (millions of CpG sites) simultaneously. Using mSTARR-seq, we identify thousands of MD regulatory elements in the human genome. MD activity is partially predictable using sequence and chromatin state information, and distinct transcription factors are associated with higher activity in unmethylated versus methylated DNA. Further, pioneer TFs linked to higher activity in the methylated state appear to drive demethylation of experimentally methylated sites. MD regulatory elements also predict methylation-gene expression relationships across individuals, where they are 1.6x enriched among sites with strong negative correlations. mSTARR-seq thus provides a map of MD regulatory activity in the human genome and facilitates interpretation of differential methylation studies.

DOI: https://doi.org/10.7554/eLife.37513.001

## Introduction

DNA methylation—the covalent addition of methyl groups to nucleotide bases, most often at CpG motifs—is a gene regulatory mechanism that plays a fundamental role in development (*Smith and Meissner, 2013*), disease susceptibility (*El-Maarri, 2005*; *Heyn and Esteller, 2012*), and the response to environmental conditions (*Jirtle and Skinner, 2007*; *Feil and Fraga, 2012*; *Joshi et al., 2018*). These observations argue that between-individual variation in DNA methylation should be important in explaining trait variation. In support of this argument, epigenome-wide association studies (EWAS) have identified thousands of correlations between DNA methylation levels at individual CpG sites and age (*Winnefeld and Lyko, 2012*; *Day et al., 2013*; *Marioni et al., 2015*), short-term (*Barrès et al., 2012*; *Pacis et al., 2015*) and long-term (*Dominguez-Salas et al., 2014*; *Tobi et al., 2014*) reactions to environmental perturbations, and disease, including cancer

(*Hansen et al., 2011*; *Hinoue et al., 2012*; *Aran et al., 2013*), type II diabetes (*Dayeh et al., 2014*), and Alzheimer's disease (*Bakulskia et al., 2012*; *De Jager et al., 2014*).

If differential DNA methylation is mechanistically important in trait variation, it should also have downstream consequences for gene regulation. However, while a functional relationship between differential methylation and gene expression levels is often assumed, experimental studies have shown that it does not always hold. For example, targeted demethylation of CpG sites within the human *HBB* promoter using TALE-TET1 fusion proteins revealed that demethylation of some CpG sites causally alters gene transcription (*Maeder et al., 2013*). At other CpG sites in the same promoter, however, demethylation had no effect on *HBB* expression. Mapping methylation-dependent (MD) regulatory activity on a genome-wide scale is therefore essential for both sifting through the growing number of DNA methylation-trait associations and for understanding the basic biology of epigenetic gene regulation. However, current approaches for assaying MD activity are limited, as they either do not provide locus-specific information (*Christman, 2002*) or can only assay a single locus per experiment (*Rivenbark et al., 2012*; *Maeder et al., 2013*; *Liu et al., 2016*) (*Table 1*). Large-scale analyses to date have thus been restricted to in vitro assays performed on sheared or synthesized DNA (*Hu et al., 2013*; *Mann et al., 2013*; *O'Malley et al., 2016*; *Zuo et al., 2017*). These studies suggest widespread differential TF sensitivity to DNA methylation levels (*Hu et al., 2013*; *O'Malley et al., 2016*; *Yin et al., 2017*; *Zuo et al., 2017*), but leave open whether and to what degree differential sensitivity translates to differences in gene expression itself.

To address this limitation, we developed a new method, 'methyl-STARR-seq' (mSTARR-seq), that can be used to assay the causal relationship between initial differences in DNA methylation and subsequent regulatory activity in high-throughput, within the cellular environment. mSTARR-seq relies on a newly engineered reporter vector, similar in setup to those used in other massively parallel reporter assays (*Arnold et al., 2013*; *Arnold et al., 2014*; *Vanhille et al., 2015*; *Vockley et al., 2015*; *Vockley et al., 2016*), but designed to eliminate all CpG sites and potential targets of bacterial methylation. mSTARR-seq then combines high-throughput cloning of hundreds of thousands of query fragments (containing millions of CpG sites) with enzymatic manipulation of DNA methylation using the enzyme *M.SssI* (*Figure 1*). Using this novel approach, we map MD regulatory activity across the human genome in the K562 immortalized human cell line. Further, we test whether regions with MD activity can be reliably predicted using information about CpG site density, endogenous chromatin state, and TF binding, and we identify both known and candidate novel TFs involved in DNA methylation-mediated gene regulation and active demethylation. Finally, we demonstrate the relevance of mSTARR-seq to data from real populations by showing that inter-individual variation in DNA methylation is more tightly coupled to inter-individual variation in gene expression for CpG sites within MD regulatory elements than sites in methylation-insensitive (non-MD) regulatory elements. Our findings indicate that a minority of CpG sites in the genome have MD regulatory activity. Thus, the functional importance of DNA methylation for gene expression at any given site should be tested empirically and not assumed as a default.

## Results

### Description of the mSTARR-seq method

mSTARR-seq is inspired by the self-transcribing active regulatory region sequencing (STARR-seq) protocol developed by *Arnold et al. (2013)*. However, the original STARR-seq vector is not appropriate for isolating the effects of DNA methylation on gene expression because it contains over 300 CpG sites in the backbone, as well as potential targets of bacterial methylation. To eliminate these confounds, we therefore engineered a novel CpG-free vector (*pmSTARRseq1*) that eliminates the possibility of bacterial *Dam* or *Dcm*-mediated methylation (*Figure 1A*; Materials and methods). As in STARR-seq, the *pmSTARRseq1* vector enables a diverse library of query fragments to be inserted in the 3' untranslated region of a constitutively expressed reporter gene, such that fragments with regulatory activity drive their own transcription when transfected into a cell type of interest (*Arnold et al., 2013*). Prior to transfection, an aliquot of the plasmid input library is treated with the methyltransferase *M.SssI*, which methylates all CpG sites (following the single locus method of (*Klug and Rehli, 2006*)). A second aliquot is treated only with *M.SssI* buffer, producing a sham treatment in which all query fragment CpG sites are left completely unmethylated. The regulatory activity

**Table 1.** Current methods for testing the causal relationship between DNA methylation and gene regulation.

| Approach | Method | Throughput | Output |
|---|---|---|---|
| *Episomal reporter* | mSTARR-seq | Genome-scale ($10^5$–$10^6$ query fragments) | Expression |
| | Luciferase reporter assay (**Klug and Rehli, 2006**) | Single locus | Expression |
| *Endogenous editing* | TALE fusion to TET1 or DNMT3a effector domains (**Maeder et al., 2013**; **Bernstein et al., 2015**) | Single locus | Expression |
| | CRISPR-dCas9 fusion to TET1 or DNMT3a effector domains (**Liu et al., 2016**; **Vojta et al., 2016**) | Single locus | Expression |
| | ZF domain fusion to DMNT3a or DNA glyocosylase effector (**Rivenbark et al., 2012**) | Single locus | Expression |
| | Inducible overexpression of an artificial ZF-DNMT3A fusion protein (**Ford et al., 2017**)[*] | Genome-scale ($10^4$ query fragments) | Expression |
| *In vitro TF binding* | Electrophoretic mobility shift assay | Single locus | TF binding |
| | Protein-binding microarray (**Mann et al., 2013**) | Genome-scale ($10^4$–$10^5$ query fragments) | TF binding |
| | DNA affinity purification sequencing (DAP-seq) (**O'Malley et al., 2016**) | Genome-scale ($10^5$–$10^6$ query fragments) | TF binding |
| | Methylation sensitive and bisulfite-SELEX (**Yin et al., 2017**) | Genome-scale ($10^4$ query fragments) | TF binding |
| | Methyl-Spec-seq (**Zuo et al., 2017**) | Genome-scale ($10^2$–$10^3$ query fragments) | TF binding |

[*]This method is currently limited to testing off-target effects in the MCF-7 cell line, which contains an inducible artificial ZF-DNMT3A fusion protein.

DOI: https://doi.org/10.7554/eLife.37513.002

of unmethylated versus methylated fragments can then be compared using high-throughput sequencing to quantify their relative abundances in reporter gene-derived mRNA. In total, this approach allows us to obtain a functional, quantitative readout of enhancer activity across the genome, and to compare the activity levels of the same regulatory element in a methylated versus unmethylated state.

To illustrate this approach, we generated an mSTARR-seq input library from randomly sheared DNA from the human GM12878 cell line, combined with additional GM12878 *MspI*-digested genomic DNA to enrich for CpG-containing fragments (*Figure 1B*; this input library contained 93% of expected fragments from a full *MspI* digest of the human genome). We then transfected

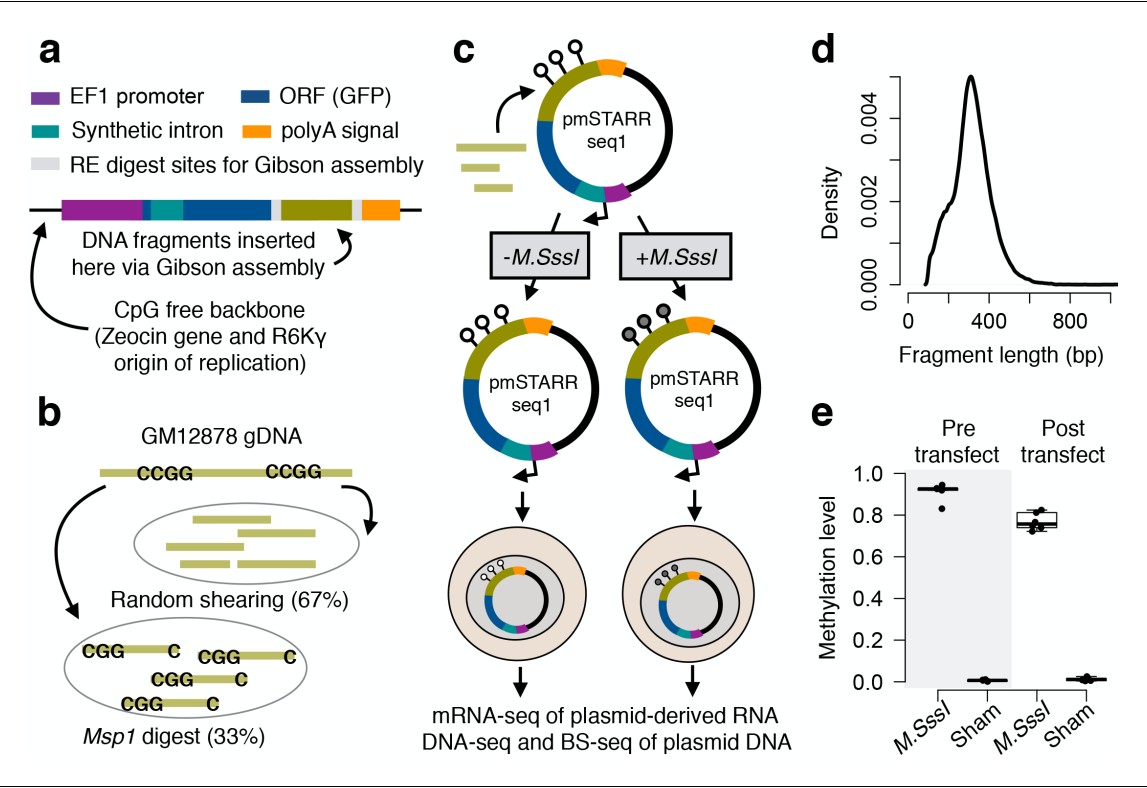

**Figure 1.** mSTARR-seq experimental design. (A) The CpG-free *pmSTARRseq1* vector is designed so that functional regulatory elements self-transcribe to produce a processed mRNA transcript, including a transcribed region (dark blue) that spans a synthetic intron (teal) and the sequence of the regulatory element itself (green). (B) GM12878 DNA was fragmented through random shearing or *Msp1* digest, which recognizes the sequence CCGG. The resulting fragments were mixed in a 2:1 ratio, and (C) cloned into *pmSTARRseq1* in high-throughput. We subjected the resulting library to either experimental methylation (*M.SssI* treatment) or a sham treatment, and transfected each pool into the K562 cell line (n = 6 replicates per condition). After a 48 hr incubation period, plasmid DNA and plasmid-derived mRNA were extracted and the variable insert regions were sequenced. (D) Inserts were of mean size 321 bp ± 107 bp s.d. (E) Bisulfite sequencing of pre- and post-transfection plasmid DNA confirmed that *M.SssI* treatment successfully methylates CpG sites introduced into *pmSTARRseq1* and that methylation status is not substantially perturbed by transfection (y-axis: mean CpG methylation level per experimental replicate, based on CpG sites in the region used for Gibson assembly, and therefore present on every plasmid). Whiskers on boxplots represent the values for the third and first quartiles, plus or minus 1.5 × the interquartile range, respectively. Evidence for lack of confounding by a transfection-induced type I interferon response is shown in *Figure 1—figure supplement 1*. An analysis of fragment diversity levels in mSTARR-seq DNA-seq and RNA-seq libraries is shown in *Figure 1—figure supplement 2*. A comparison of fragment diversity between this experiment and other high-throughput reporter assays, as well as the effect of repeated library transformation on diversity is shown in *Figure 1—figure supplement 3*. mSTARR-seq coverage by genomic compartment and ENCODE chromatin state annotations is shown in *Figure 1—figure supplement 4*.

DOI: https://doi.org/10.7554/eLife.37513.003

The following figure supplements are available for figure 1:

**Figure supplement 1.** Plasmid transfection does not induce a strong type I interferon (IFN-I) response.

DOI: https://doi.org/10.7554/eLife.37513.004

**Figure supplement 2.** Diversity in plasmid DNA-seq libraries versus mRNA-seq libraries.

DOI: https://doi.org/10.7554/eLife.37513.005

**Figure supplement 3.** Fragment diversity in mSTARR-seq libraries is comparable to previous work, and this diversity can be easily regenerated across experiments.

DOI: https://doi.org/10.7554/eLife.37513.006

**Figure supplement 4.** Regions covered by mSTARR-seq.

DOI: https://doi.org/10.7554/eLife.37513.007

unmethylated and methylated versions of the plasmid library into the K562 myeloid lineage cell line (*Figure 1C*; we performed six replicate transfections per condition, giving us sample sizes that were similar to or exceeded previous STARR-seq (*Arnold et al., 2013*; *Arnold et al., 2014*; *Zabidi et al., 2015*) and MPRA (*Melnikov et al., 2012*; *Patwardhan et al., 2012*; *Tewhey et al., 2016*)

experiments). Forty-eight hours post-transfection, we isolated and sequenced both the plasmid-derived mRNA and the fragment inserts from each of the 12 replicates (*Figure 1D*; *Figure 1—figure supplement 1*; *Supplementary file 1*). We confirmed that transfection itself did not disrupt the expected DNA methylation state by performing bisulfite sequencing on the plasmid DNA and estimating methylation levels from 2 CpG sites introduced from Illumina-style adapters ligated to the human DNA inserts (*Figure 1E*). We also confirmed that our results were not confounded by two issues recently raised for classical STARR-seq assays (*Muerdter et al., 2018*), namely: (i) enhancer interactions with the bacterial plasmid origin-of-replication, which contains core promoter sequence, instead of the intended core promoter (see Materials and methods), and (ii) induction of a type I interferon (IFN) response during DNA transfection (note that K562s do not express cGAS/STING pathway genes and are thought to be unaffected by this issue (*Muerdter et al., 2018*)). To confirm that K562s do not mount an IFN response to mSTARR-seq plasmid transfection, we measured the expression levels of six genes identified by Muerdter and colleagues as diagnostic of IFN activity. Expression levels of 5/6 genes were not affected by transfection (p>0.05); one gene, *OAS3* did increase, but this increase was not significant after correction for multiple hypothesis testing (*Figure 1—figure supplement 1*). Additionally, we found that regions with significant mSTARR-seq activity were not enriched for IFN-1 signaling or other immune-related pathways (*Supplementary file 2*).

Using mSTARR-seq, we assayed ~750,000 unique DNA fragments (mean ± SD = 759,725±252,187 fragments per replicate; *Figure 1D*), comparable to or exceeding the diversity in published STARR-seq and massively parallel reporter assay experiments (*Figure 1—figure supplements 2* and *3*). For subsequent analysis, we binned the genome into 200 bp non-overlapping intervals and filtered these regions to focus on the 262,829 intervals that overlapped at least one mSTARR-seq mRNA read and one DNA read in at least half of the replicates in each condition (see Materials and methods). The resulting data set covered 1.81 million unique CpG sites, 57% of fragments expected from a complete *MspI* digest of the human genome, and 13% of the K562 euchromatic genome (*Figure 2A*), (*Figure 1—figure supplement 4*). Within this data set, we identified 19,703 intervals with an excess of plasmid-derived mRNA relative to DNA input in at least one condition (unmethylated or methylated), which indicates regulatory/enhancer activity (8% of analyzed regions; linear model, FDR < 10%: *Supplementary file 3*). Information on dynamic range, as well as how the number of assayed fragments, the number of analyzed CpG sites, and the coverage of CpG sites scales with sequencing depth is provided in *Figure 2—figure supplements 1* and *2*.

Among regions with evidence for enhancer activity, 2969 exhibited significant methylation-dependent (MD) activity (15% of analyzed regions; linear model, FDR < 10%). Eighty-six percent of these MD enhancers were more active when unmethylated and 14% were more active when methylated (*Figure 3A*; *Supplementary file 4*). Only 4 of the 325 CpG-free regions in the analysis set were inferred to have MD activity, consistent with an overall low false discovery rate (*Figure 3B*).

## Validation against low-throughput reporter assays and comparison with endogenous measures of regulatory function

To validate mSTARR-seq's performance, we compared it to luciferase reporter assays designed to test for regulatory activity or DNA methylation-dependent regulatory activity at a single locus (*Klug and Rehli, 2006*). We assayed 18 candidate regulatory elements spanning a range of enhancer activities and MD enhancer activities inferred from our mSTARR-seq experiments (including loci with no detectable MD enhancer activity; *Supplementary file 5*). In both cases, we observed strong agreement between the conventional low-throughput method and results from mSTARR-seq: enhancer activity inferred from unmethylated plasmids was strongly correlated between mSTARR-seq and single locus assays (linear model, $R^2$ = 0.86, p < $10^{-15}$, *Figure 2C*), as was MD enhancer activity (based on comparisons between unmethylated and methylated plasmids; linear model, $R^2$ = 0.76, p < $10^{-15}$, *Figure 2C*). These results indicate that mSTARR-seq reliably scales previous methods for assessing MD regulatory activity—in this case, by five orders of magnitude—without loss of reliability.

To further validate mSTARR-seq, we also investigated concordance between our results and endogenous enhancer annotations for K562 cells. As expected, the set of regions capable of enhancer activity was highly enriched for 'weak enhancer', 'strong enhancer', and 'active promoter' ENCODE chromatin states (*Dunham et al., 2012*) (two-sided Fisher's exact test, $\log_2$(odds) = 0.219–

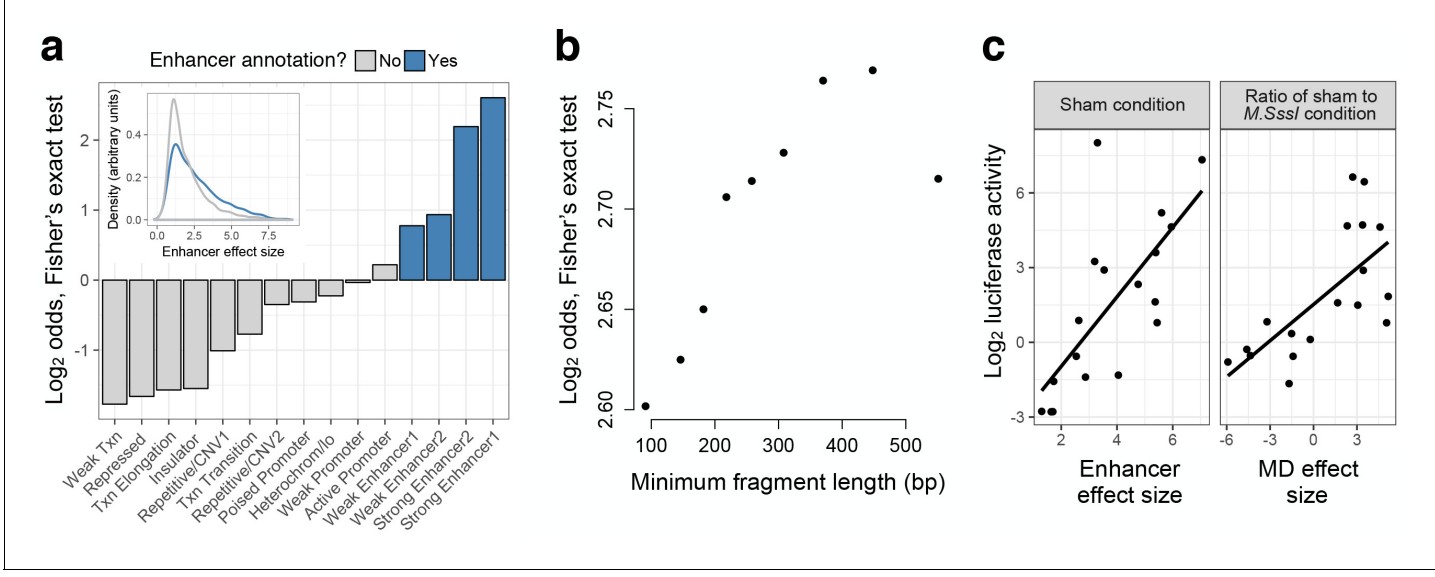

**Figure 2.** mSTARR-seq identifies regions with endogenous regulatory activity. (**A**) Regions with significant regulatory activity are enriched for enhancer chromatin states in K562 cells (blue). Y-axis shows the two-sided Fisher's Exact Test log2(odds) for enrichment/depletion of mSTARR-seq enhancers in 15 K562-annotated chromatin states (p<0.05 for all tests except 'Weak promoter'). Inset: mSTARR-seq effect size for regions in enhancer chromatin states versus other chromatin states, for regions with significant activity only. (**B**) Binning regions with significant mSTARR-seq enhancer activity by fragment length reveals that larger fragments are more strongly enriched for ENCODE-annotated 'strong enhancers'. The y-axis depicts the log2(odds) from a two-sided Fisher's exact test for enrichment of mSTARR-seq enhancers (binned by deciles of fragment length) in either of the two 'strong enhancer' chromatin states (p<0.05 for all tests). (**C**) Agreement between conventional CpG-free luciferase reporter assays (***Klug and Rehli, 2006***) and enhancer activity (left) and MD-dependent activity (right) estimated from mSTARR-seq for 18 candidate regulatory elements (***Supplementary file 3***). mSTARR-seq activity explains 86.0% and 76.1% (p < $10^{-15}$) of variance in normalized luciferase activity, respectively (linear mixed model controlling for batch and assay fragment length). The dynamic range for mSTARR-seq enhancer detection is shown in ***Figure 2—figure supplement 1***. The effects of sample size and sequencing effort on coverage, regions analyzed, and CpG sites analyzed are shown in ***Figure 2—figure supplement 2***. Sensitivity versus specificity of enhancer detection as a function of fragment length is shown in ***Figure 2—figure supplement 3***.

DOI: https://doi.org/10.7554/eLife.37513.008

The following figure supplements are available for figure 2:

**Figure supplement 1.** Effect size distributions for regions identified as enhancers (FDR < 0.1).
DOI: https://doi.org/10.7554/eLife.37513.009
**Figure supplement 2.** Effect of sequencing depth and sample size on the data set properties.
DOI: https://doi.org/10.7554/eLife.37513.010
**Figure supplement 3.** Known enhancers on larger DNA query fragments are more likely to exhibit significant regulatory activity in mSTARR-seq.
DOI: https://doi.org/10.7554/eLife.37513.011

2.60, all p < $10^{-7}$; ***Figure 2A***), and regions annotated as 'strong enhancer' contained the largest proportion of mSTARR-seq-identified regulatory elements (31% of those tested). Further, regions with mSTARR-seq enhancer activity that fell in enhancer or active promoter chromatin states consistently displayed larger effect sizes than those found in other chromatin states (linear model, $R^2$ = 0.11, p < $10^{-15}$; ***Figure 2A***). Hence, consistent with findings from previous massively parallel reporter assays (***Arnold et al., 2013***), mSTARR-seq can identify both endogenously active regulatory elements and additional loci with latent regulatory potential. Notably, power to detect enhancer activity increased with larger query fragment sizes (***Figure 2B***) and the trade-off between sensitivity and specificity also improved (***Figure 2—figure supplement 3***), possibly because short fragments eliminate binding sites key to enhancer activity. Thus, while we used a mean fragment size typical for STARR-seq experiments in this analysis (***Arnold et al., 2013***; ***Arnold et al., 2014***; ***Zabidi et al., 2015***) (but larger than most synthesized fragments in MPRA experiments: (***Melnikov et al., 2012***; ***Patwardhan et al., 2012***; ***Tewhey et al., 2016***)), additional increases in fragment size would likely further increase assay sensitivity. Indeed, our results indicate that the current data set is more sensitive to strong enhancers than weak enhancers (***Figure 3A***), and that we likely miss some true weak

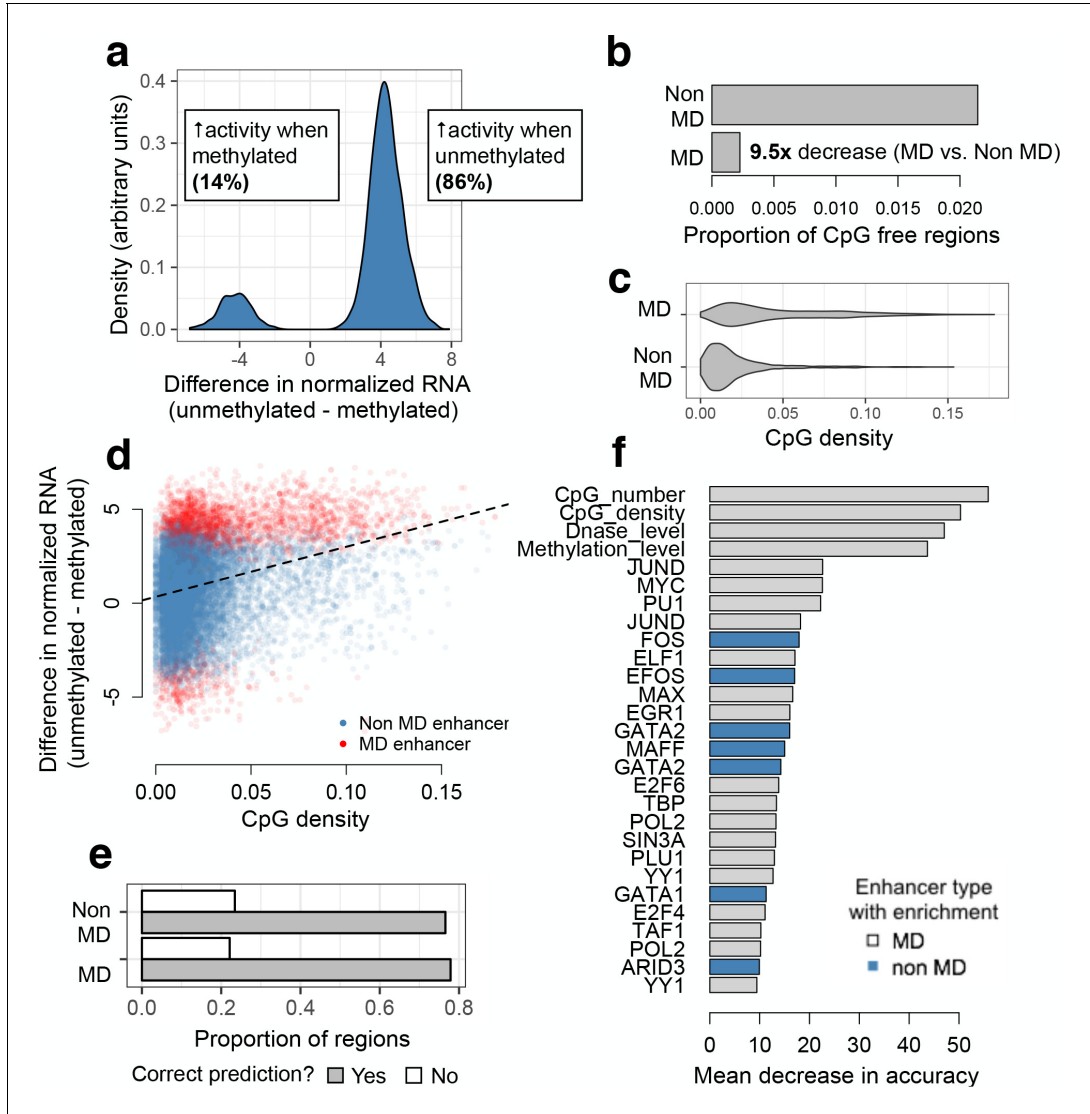

**Figure 3.** mSTARR-seq identification and prediction of MD enhancers. (**A**) The distribution of differences in normalized mRNA transcript abundance between the unmethylated and methylated conditions for all significant MD enhancers. (**B**) CpG-free MD enhancers occur at a 9.5-fold lower rate than CpG-free windows with no MD enhancer activity. (**C**) Distribution of fragment CpG density for regions identified as MD versus non-MD enhancers. (**D**) CpG-dense mSTARR-seq enhancers tend to be repressed by DNA methylation (positive y-axis value; Spearman's rho = 0.284, p<10^{-15}; n = 19,703 regions with mSTARR-seq regulatory activity). (**E**) The proportion of non-MD and MD enhancers that were accurately classified via a random forests (RF) classifier. (**F**) Features that distinguish MD and non-MD enhancers in the RF classifier (10% FDR; note that the feature set [***Supplementary file 6***] includes some repeated ChIP-seq experiments for the same TF). X-axis: mean decrease in predictive accuracy when permuting the focal variable. Blue: positive prediction of non-MD enhancers; gray: positive prediction of MD enhancers. Mean decrease in accuracy is calculated as the mean difference in accuracy between 1000 models in which the focal variable is permuted and one where it is not, normalized by the standard deviation of these difference values; it is thus analogous to a standardized effect size. Statistical calibration and dynamic range of detection for models used to detect MD activity are shown in ***Figure 3—figure supplement 1***. Effect of varying the threshold used to predict MD versus non-MD enhancers is shown in ***Figure 3—figure supplement 2***. Motifs for TFs that are significant predictors of MD enhancer activity are shown in ***Figure 3—figure supplement 3***.
DOI: https://doi.org/10.7554/eLife.37513.012

The following figure supplements are available for figure 3:

**Figure supplement 1.** Linear models for detecting MD enhancers are well-calibrated, but primarily detect large effect size regions.
DOI: https://doi.org/10.7554/eLife.37513.013

**Figure supplement 2.** Impact of the threshold used to assign enhancers to the MD or non-MD class in random forests analyses.
DOI: https://doi.org/10.7554/eLife.37513.014

**Figure supplement 3.** Motifs for transcription factors identified as predictors of MD regulatory activity in random forest analyses.
DOI: https://doi.org/10.7554/eLife.37513.015

and/or weakly MD enhancers, after FDR correction for multiple testing (*Figure 2—figure supplement 1*; *Figure 3—figure supplement 1*).

## Determinants of methylation-dependent (MD) regulatory activity

Our results allowed us to investigate the properties that predict MD regulatory activity and the transcription factors associated with MD enhancers. Overall, we found that MD enhancers have higher CpG densities and contain more CpG sites than non-MD enhancers (two-sided Wilcoxon-signed rank test, $p < 10^{-15}$; *Figure 3C*). However, CpG density explained only a small amount of variation in the magnitude of methylation dependence, suggesting that other characteristics also contribute to quantitative variation in MD activity (Spearman's rho = 0.284, $p < 10^{-15}$; *Figure 3D*).

To identify these characteristics, we used a random forests (RF) classifier to evaluate the contribution of 117 genomic features to differentiating MD and non-MD enhancers. Specifically, we compared the 2553 regions suppressed by methylation to a set of 2503 regions with strong evidence against methylation-dependence (i.e. >50% FDR in our analysis of MD activity). Our feature set included information about absolute CpG number and density; endogenous chromatin state, chromatin accessibility, and DNA methylation levels (*Dunham et al., 2012*; *Kundaje et al., 2015*); evolutionary conservation (*Siepel et al., 2005*); and TF binding from K562 ENCODE ChIP-seq data (*Dunham et al., 2012*). The resulting RF model predicted MD regulatory element activity with 78.8% accuracy (72.3% without the top features of CpG density and CpG number; *Figure 3E* and *Figure 3—figure supplement 2*). In addition to CpG number and density, 26 features were identified as significant predictors (permutation-based (*Archer, 2015*) FDR < 10%; *Figure 3F* and *Supplementary file 6*). Measures of DNase I hypersensitivity and endogenous methylation levels were among the top predictive features: relative to non-MD enhancers, enhancers suppressed by DNA methylation were more likely to occur in open chromatin or in highly methylated regions of the genome. MD enhancers were also more likely to contain binding sites for TFs that contain CpG sites in their canonical binding motifs (*Figure 3F* and *Figure 3—figure supplement 3*).

Consistent with these results, many TFs are thought to be sensitive to DNA methylation levels in or near their binding motifs (*Hu et al., 2013*; *O'Malley et al., 2016*; *Yin et al., 2017*). This ability to 'read' epigenetic modifications to DNA sequence could contribute to variation in MD regulatory activity in our data set. To test this possibility, we therefore investigated TF binding motif enrichment within mSTARR-seq identified MD enhancers, using motifs defined in the HOMER database (*Heinz et al., 2010*). Among the 2543 MD enhancers in which DNA methylation suppresses activity, we identified 32 significantly enriched TF-binding motifs (relative to all regions with mSTARR-seq regulatory activity, FDR < 1%; *Figure 4A* and *Supplementary file 7*). These results agree with known cases of methylation-dependent binding for well-studied classes of TFs (e.g. ETS family TFs, some of which are known to be methylation-sensitive (*Polansky et al., 2010*; *Cooper et al., 2015*; *Stephens and Poon, 2016*)). Additionally, they are significantly correlated with estimates of methylation-dependent TF binding to naked DNA using a completely orthogonal method (*Yin et al., 2017*). Specifically, when we compared estimates of motif enrichment from mSTARR-seq-identified MD enhancers versus estimates of affinity for methylated DNA in SELEX experiments, we found significant agreement for both TFs that preferentially bound unmethylated (linear model, $R^2 = 0.342$; $p = 1.33 \times 10^{-6}$) and TFs that preferentially bound methylated DNA (linear model, $R^2 = 0.113$, $p = 0.028$; *Figure 5A–B*).

Among the 13 TF motifs associated with *increased* enhancer activity when DNA was methylated, we identified a strong overrepresentation of TFs from the GATA subfamily of zinc finger TFs (15.9x enrichment, hypergeometric test $p=1.74\times10^{-8}$; *Figure 4B* and *Supplementary file 8*). This observation is consistent with reports that GATA3 and GATA4 can bind to methylated DNA outside the cellular context (*Hu et al., 2013*). We therefore compared our findings to published chromatin accessibility data for wild-type murine stem cells, which contain normal patterns of DNA methylation, and triple knockouts for *DNMT1*, *DNMT3a*, and *DNMT3b*, in which DNA methylation is abolished (*Domcke et al., 2015*). For five of ten tested GATA family TFs, open chromatin regions specific to wild type (i.e. those absent in the triple knockouts) were significantly enriched for their cognate-binding sites (*Figure 4C*), in support of the idea that GATA family TFs preferentially bind methylated DNA as part of their 'pioneer' factor activity (*Zhu et al., 2016*). In contrast, ETS family TF binding sites were almost universally (38 of 41 tested) enriched in DNMT knockout-specific open chromatin regions.

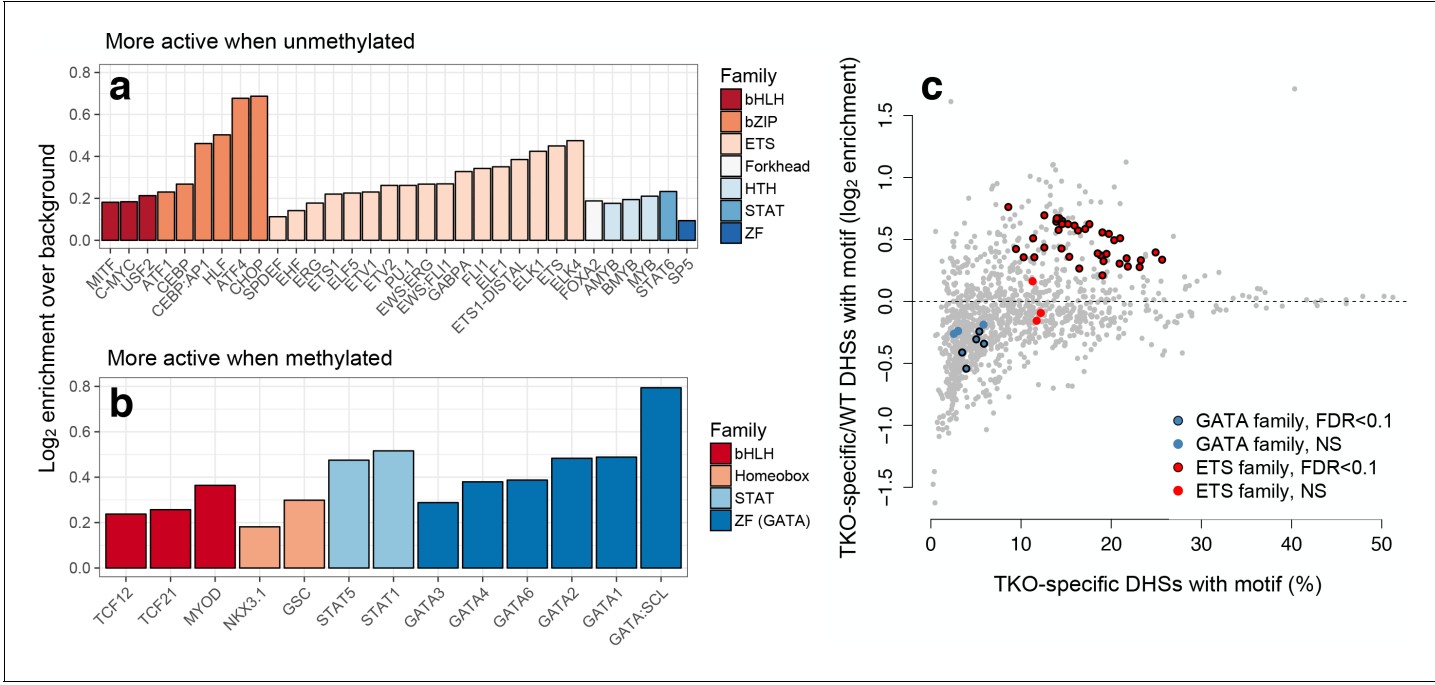

**Figure 4.** mSTARR-seq identifies MD transcription factor-DNA binding. (**A**) Transcription factor motifs enriched in MD enhancers that are more active when unmethylated, colored by TF family. (**B**) TF motifs enriched in MD enhancers that are more active when methylated. (**C**) DNase hypersensitive sites (DHS) specific to murine stem cells that lack DNA methylation (DNMT triple knock-outs: TKO) are strongly enriched for ETS family binding sites relative to wild-type cells with intact DNA methylation. In contrast, DHSs specific to wild-type cells are enriched for GATA family binding sites relative to triple knock-outs. DHS data are from[22]. X-axis: % knockout-specific DHSs that contain a given TF-binding motif (n = 1251 motifs). Y-axis: Ratio of knockout versus wild-type specific DHSs containing a given TF-binding site motif. Colored dots circled in black show significant enrichment for an ETS or GATA family TF (10% FDR, hypergeometric test). Evidence for CpG site and TFBS-dependent convergence to endogenous K562 methylation levels for some tested sites is shown in *Figure 4—figure supplement 1*.

DOI: https://doi.org/10.7554/eLife.37513.016
The following figure supplement is available for figure 4:

**Figure supplement 1.** Methylation levels of human insert fragments in mSTARR-seq experiments converge toward endogenous methylation patterns post-transfection.
DOI: https://doi.org/10.7554/eLife.37513.017

Because the treatment condition—methylated versus unmethylated—is the only exogenous variable in mSTARR-seq, all signals of MD activity we detect depend on the original treatment status. In this respect, mSTARR-seq is conservative: rapid demethylation of the methylated treatment or de novo methylation of the unmethylated treatment would attenuate or eliminate any MD signal, as the two treatments would effectively become identical. However, methylation-dependent TF binding could also lead to subsequent changes in DNA methylation, as some TFs are associated with demethylating activity (*Suzuki et al., 2017*). To test this possibility, we generated an additional locus-specific bisulfite sequencing data set on fragments extracted 48 hr post-transfection. As in our previous experiments (*Figure 1E*), introduced CpG sites in the fragment-backbone linker retained their experimentally-induced methylation status (mean CpG methylation levels = 1.6% and 95.1% in the unmethylated and methylated condition, respectively). However, CpG sites in fragments that mapped to the human genome partially converged towards endogenous levels in K562s, for both unmethylated (rho = 0.476, $p<10^{-16}$) and methylated replicates (rho = 0.223, $p<10^{-16}$; *Figure 4—figure supplement 1*). Specifically, 23.1% of sites in the unmethylated condition became methylated, while 15.7% of sites in the methylated condition lost methyl marks.

This effect was driven in part by altered methylation levels at specific transcription factor binding sites. We observed the strongest evidence for post-transfection demethylation at K562 ChIP-seq-identified binding sites for ZNF274 and GATA2 (loss of a mean of 35.9% and 26.5% methylation, respectively), and for post-transfection de novo methylation at ChIP-seq-identified binding sites for

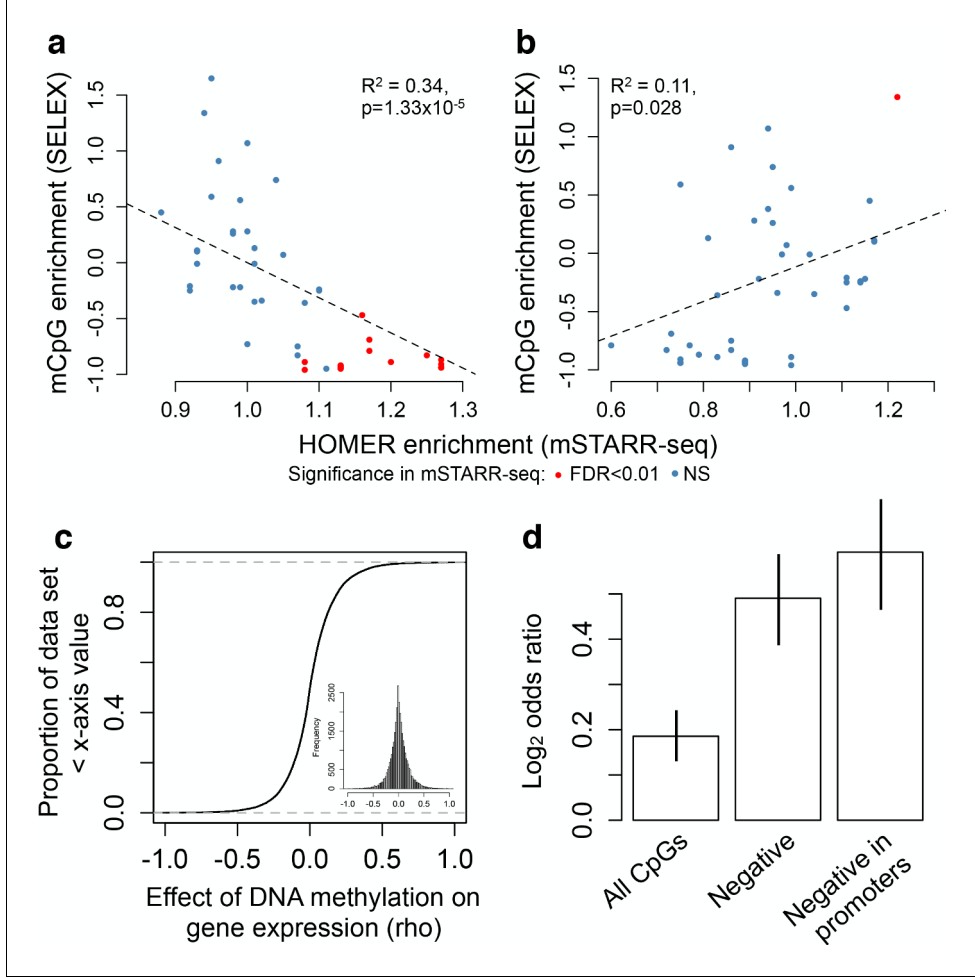

**Figure 5.** Agreement with in vitro and in vivo estimates of methylation-dependence. (A–B) Comparison of motif enrichment for mSTARR-seq-identified MD enhancers (x-axis) versus affinity for methylated DNA in SELEX experiments (y-axis: higher values represent preferential binding to methylated DNA). Results are plotted for each TF tested in both mSTARR-seq and in *Yin et al. (2017)* (A) TFs enriched for MD enhancers that were more active when unmethylated. (B) TFs enriched for MD enhancers that were more active when methylated. Each TF is colored by whether its binding motif was significantly enriched in the given MD enhancer set in mSTARR-seq. (C) CDF of the effect of CpG methylation levels on gene expression levels (rho) for 32,843 gene pairs measured in 1202 human monocyte samples (*Reynolds et al., 2014*). The mean correlation between CpG methylation levels and the expression of the closest gene is near zero. Inset shows the same distribution plotted as a histogram. (D) Enrichment ($Log_2$ odds ratio from a Fisher's exact test) of CpG sites with significant DNA methylation-gene expression correlations (FDR < 10%) in MD enhancers relative to non-MD enhancers. Bars show (i) enrichment versus all CpG sites measured in both mSTARR-seq and the monocyte dataset; (ii) enrichment in the subset of CpG sites with negative DNA methylation-gene expression correlations (of any magnitude); and (iii) enrichment in the subset of CpG sites with negative DNA methylation-gene expression correlations located in gene promoters. Error bars represent 95% confidence intervals. Evidence that major environmental perturbations to gene expression (IFNA challenge) leads to correlated changes in DNA methylation is shown in *Figure 5—figure supplement 1*.

DOI: https://doi.org/10.7554/eLife.37513.018

The following figure supplement is available for figure 5:

**Figure supplement 1.** IFNA treatment produces changes in enhancer activity that correlate with changes in post-transfection plasmid DNA methylation levels.
DOI: https://doi.org/10.7554/eLife.37513.019

POL2 and KAP1 (gain of a mean 26.0%, and 18.6% methylation, respectively; *Supplementary file 9*).

However, GATA family TFs showed some of the most consistent evidence for demethylation overall: among the top 20 TFs ranked by mean loss of methylation (in the methylated condition), GATA family TFs were enriched 6.7-fold relative to chance expectations ($p=3.60\times10^{-4}$). Our data are thus consistent with both methylation-dependent binding and subsequent demethylating activity at GATA-bound TFBS (*Zhu et al., 2016*; *Suzuki et al., 2017*), and support the argument that sequence alone provides important information about cell-typic DNA methylation levels (*Schübeler, 2015*). They also suggest that, while not the original motivation for the method, mSTARR-seq could be applied to screen for TFs associated with active demethylation.

## mSTARR-seq explains site-to-site heterogeneity in the strength of DNA methylation level-gene expression level correlations in vivo

Our mSTARR-seq results indicate that, at many CpG sites, DNA methylation may be functionally silent (i.e. it has no effect on regulatory activity). This result may explain the observation that, in population data sets, inter-individual variation in DNA methylation levels is a good predictor of inter-individual variation in gene expression levels at some CpG sites, but not others (*Lam et al., 2012*; *Reynolds et al., 2014*). Specifically, DNA methylation-gene expression correlations are expected to be stronger at CpG sites where DNA methylation causally affects gene expression than at sites where DNA methylation is functionally silent.

To test this prediction, we drew on paired DNA methylation and gene expression data for 1202 human primary monocyte samples (*Reynolds et al., 2014*). In this data set, the correlation between CpG methylation levels and the expression level of the closest gene is relatively strong within a single individual's genome (mean rho = -0.152 ± 0.018 s.d., rho calculated for each of n = 1202 paired genome-wide methylation and gene expression data sets, all $p<10^{-16}$), but near zero when estimated across individuals (mean rho = 0.008 ± 0.188 s.d., n = 32,843 site-gene pairs measured in 1202 individuals each; *Figure 5C*). However, in mSTARR-seq identified MD enhancers, there is an enrichment of sites with significant DNA methylation-gene expression correlations across individuals (FDR < 10%), relative to non-MD enhancers (two-sided Fisher's exact test, $\log_2$odds = 0.186, $p=4.967\times10^{-8}$). This enrichment increases further for sites in the monocyte data set that display the canonical negative correlation between DNA methylation and gene expression levels ($\log_2$odds = 0.491, $p<10^{-16}$), and is strongest when these sites also occur in promoter regions ($\log_2$odds = 0.592, $p<10^{-16}$; *Figure 5D*). Overall, CpG sites found in mSTARR-seq MD regulatory elements are 1.6-fold more common among sites where DNA methylation is strongly negatively correlated with gene expression (rho $<-0.2$, FDR < 10%) than among all other sites.

mSTARR-seq data can therefore help discriminate between sites where population variation in DNA methylation can drive differences in gene expression, and those where differential methylation is either independent or downstream of gene expression variation. Notably, although mSTARR-seq is designed to test the effects of experimentally induced DNA methylation on gene expression, it is consistent with the observation that major environmental or developmental perturbations to gene expression are also associated with changes in DNA methylation. To demonstrate this property, we challenged K562s carrying methylated mSTARR-seq constructs with interferon alpha (IFNA), which induces a strong antiviral innate immune response. We identified 7010 CpG sites that occurred within 2678 regions tested for enhancer activity in both the control and IFNA-challenged samples, and for which we were also able to measure IFNA-induced changes in DNA methylation levels. Increased enhancer activity post-IFNA treatment was significantly correlated with loss of DNA methylation at individual sites (rho = 0.043, p=0.025; n = 7010 independently analyzed CpG sites), but not significantly correlated with mean loss of CpG methylation across each region (n = 2678, 200 bp regions; rho = 0.014, p=0.687). For both region- and site-level analyses, the strength of the correlation between IFNA induction and loss of DNA methylation became progressively stronger for more strongly induced enhancers (up to rho = 0.125 or rho = 0.139 for the 15th quantile of induced enhancers, for regions and sites, respectively, p<0.05 for both; *Figure 5—figure supplement 1*). The magnitude of the IFNA effect on enhancer activity thus was a significant predictor of CpG demethylation versus maintenance (logistic regression: p=0.043, $R^2$ = 0.071, n = 7010 CpG sites).

## Discussion

Here, we present a novel high-throughput assay for performing genome-scale tests of methylation-dependent regulatory activity. Our results demonstrate the utility of mSTARR-seq for differentiating between regions of the genome in which DNA methylation is causally important for gene regulation and those in which it is effectively silent. mSTARR-seq thus provides an efficient, cost-effective strategy for interrogating the functional importance of candidate genomic regions. Notably, the information obtained from the original STARR-seq protocol (*Arnold et al., 2013*), which tests a sequence's capacity to act as an enhancer, is also generated as a by-product of the mSTARR-seq protocol (e.g. data from the unmethylated condition is analogous to STARR-seq data, although the vector backbone differs).

Together, our findings emphasize substantial variability in the functional relationship between DNA methylation and gene regulation across the genome. In particular, our data indicate that MD activity is probably less common for CpG sites in the human genome than methylation-independent activity, at least in any given cell type and environmental condition; further, where it does occur but was missed by our analysis, dependence on DNA methylation is likely to be weak. In addition, in agreement with several proposed models for the role of DNA methylation in gene regulation, we find that both basic sequence characteristics (e.g. density of CpG sites) and the sensitivity of their cognate TFs to local methylation play an important role in predicting MD elements (*Domcke et al., 2015*; *Schübeler, 2015*). Our results provide support for the hypothesis that pioneer TFs, such as members of the GATA TF family, have a higher affinity for methylated DNA, potentially aiding in their ability to bind condensed chromatin (*Zhu et al., 2016*). Indeed, methylated fragments that contained endogenous GATA-binding sites also tended to become demethylated during our experiments, consistent with the role of pioneer TFs in remodeling the epigenome. Other TFs important in development and cell fate, such as MyoD and TCF21, are also enriched among MD enhancers with increased activity when methylated, supporting the idea that preferential binding of methylated DNA could be used to aid in pioneer TF discovery (*Zhu et al., 2016*). A caveat to our approach is that, like all massively parallel reporter assays, the regulatory elements we identify operate outside the context of the endogenous genome (and thus, interactions between DNA methylation and higher order chromatin structure will be missed). Consequently, mSTARR-seq isolates the effects of DNA methylation alone, identifying the subset of MD elements for which changes in DNA methylation are sufficient to causally alter gene regulation.

Finally, mSTARR-seq results can be used to identify the CpG sites for which interindividual variation in DNA methylation is most tightly linked to gene expression variation in vivo. Thus, maps of MD regulatory activity obtained from mSTARR-seq can help prioritize DNA methylation-trait associations for further investigation. Notably, the effects of DNA methylation on gene expression are not only cell-type-dependent, but may also vary depending on environment, genotype, or species (*Jirtle and Skinner, 2007*; *Zhang and Meaney, 2010*; *Hernando-Herraez et al., 2013*; *Banovich et al., 2014*; *Hernando-Herraez et al., 2015*). mSTARR-seq can be adapted to investigate these dependencies by varying the transfected cell types, input DNA fragment library (e.g. via targeted capture or ChIP (*Vanhille et al., 2015*; *Vockley et al., 2016*)), or the cellular environment (e.g. through in vitro stimulation of transfected cells, similar to the IFNA treatment reported here). Thus, we expect that mSTARR-seq will become a useful tool for facilitating screening and dissection of the causal relationship between DNA methylation and gene expression levels in many contexts.

## Materials and methods

**Key resources table**

| Reagent type (species)or resource | Designation | Source or reference | Identifiers | Additional |
|---|---|---|---|---|
| Cell line (human) | K562 | ATCC | ATCC CCL-243 | |
| Recombinant DNA reagent | pmSTARRseq1 | Addgene | Plasmid #96945 | |
| Peptide, recombinant protein | IFN-α | Abcam | ab48750 | |

*Continued on next page*

*Continued*

| Reagent type (species)or resource | Designation | Source or reference | Identifiers | Additional |
|---|---|---|---|---|
| Chemically competent *E. coli* cells | GT115 strain | Invivogen | ChemiComp GT115 | |
| Electrically competent *E. coli* cells | GT115 strain | Intact Genomics | | Custom order to grow Invivogen's ChemiComp GT115 strain and prepare cells for electroporation |

### pmSTARRseq1 design

We constructed a self-transcribing CpG-free vector that was built from Invivogen's pCpGfree-pro-moter-Lucia plasmid, which contains an R6Kγ origin of replication and a Zeocin resistance gene. Spe-cifically, we replaced the region between the *NsiI* and *NheI* digest sites of the pCpGfree-promoter-Lucia plasmid with the following sequence:

ATGCATAGGGAGAAGAGCATGCTTGAGGGCTGAGTGCCCCTCAGTGGGCAGAGAGCACA
TGGCCCACAGTCCCTGAGAAGTTGGGGGGGAGGGGTGGGCAATTGAACTGGTGCCTAGAGAAGG
TGGGGCTTGGGTAAACTGGGAAAGTGATGTGGTGTACTGGCTCCACCTTTTTCCCCAGGG
TGGGGGAGAACCATATATAAGTGCAGTAGTCTCTGTGAACATTCAGGGCCTGAATTAATTCACTG
TCTGCCAGGGCCAGCTGTTGGGGTGAGTACTCCCTCTCAAAAGCTGGCATGACTTCTGCAC
TAAGATTGTCAGTTTCCAAAAATGAGGAGGATTTGATATTCAACTGGCCCTCAGTGATGCC
TTTGAGGGTGGCTGGGTCCATCTGGTCAGAAAAGACAATCTTTTTGTTGTCAAGCTTGAGGTG
TGGCAGGCTTGAGGTCTGGCCATACACTTGAGTGACAATGACATCCACTTTGCCTTTCTCTCCA-
CAGGTGTCCACTCCCAGGTCCAACTGCAGGTCACCTGCAGGCTTAAGCTCATGGTTTCTAAGG-
GAGAAGAACTCTTTACTGGTGTTGTCCCAATTCTGGTTGAGCTGGATGGTGATGTGAATGGCCA-
CAAATTCTCTGTGTCTGGTGAAGGTGAAGGAGATGCAACTTATGGAAAGCTGACTCTGAAGTTCA
TTTGTACAACAGGAAAGCTGCCAGTGCCTTGGCCAACTCTGGTGACCACCCTGACTTATGGTG
TTCAATGTTTCAGCAGGTACCCTGACCACATGAAGCAGCATGACTTCTTTAAATCTGCAATGCCA-
GAAGGTTATGTTCAGGAGAGGACAATCTTCTTTAAGGATGATGGAAATTATAAGACAAGGGCA-
GAAGTGAAGTTTGAAGGTGATACACTGGTTAACAGAATTGAGCTGAAAGGCATTGATTTTAAG-
GAAGATGGAAACATTCTGGGTCACAAGCTGGAGTACAACTATAATTCTCACAATGTTTACATTA
TGGCAGATAAGCAGAGGAATGGAATTAAGGCTAATTTCAAGATTAGACACAACATTGAGGA
TGGGTCTGTCCAACTGGCAGACCATTACCAGCAGAACACCCCTATTGGTGATGGCCCAGTTCTCC
TCCCAGATAATCACTATCTCAGCACTCAATCTGCTCTGTCCAAAGACCCTAATGAGAAAAGAGAC-
CACATGGTCCTCCTGGAGTTTGTGACAGCAGCAGGAATTACTCTGGGAATGGATGAGCTG
TACAAGGCCAAGTTGACCAGTGCTGTCCCAGTGCTCACAGCCAGGGATGTGGCTGGAGCTG
TTGAGTTCTGGACTGACAGGTTGGGGTTCTCCAGAGATTTTGTGGAGGATGACTTTGCAGGTG
TGGTCAGAGATGATGTCACCCTGTTCATCTCAGCAGTCCAGGACCAGGTGGTGCCTGACAA-
CACCCTGGCTTGGGTGTGGGTGAGAGGACTGGATGAGCTGTATGCTGAGTGGAGTGAGGTGGTC
TCCACCAACTTCAGGGATGCCAGTGGCCCTGCCATGACAGAGATTGGAGAGCAGCCC
TGGGGGAGAGAGTTTGCCCTGAGAGACCCAGCAGGCAACTGTGTGCACTTTGTGGCAGAGGAG-
CAGGACTAAAGTCTAGAGCTTGTACTAGTGGTGATTCCCCTGACGCGTGCACGTCTGCTGACGCG
TAAAGTCTCCCGTGAACTTTACCCACGCGTGCATATCGGGGATGAAAGACGCGTCATGATGAC-
CACCGATATGGCCAGTGTACGCGTCTCGGGGAATATAAATACACAGCCAGTCTGCAGGACA
TCCCATGGGAATTCAGCCAGCCACTTCAACTGCTAGC

This introduced sequence contains the following elements: a CpG-free EF1 promoter (Invivogen), a CpG-free version of a synthetic intron (pIRESpuro3, Clontech), an ORF (the CpG-free CFP::Sh gene from Invivogen), and a cluster of *MluI* digest sites for screening. The *MluI* cluster is flanked by restric-tion enzyme digest sites for *SpeI* and *NcoI* to enable high-throughput replacement of this sequence with candidate regulatory elements. To avoid any potential for bacterial methylation, we designed the plasmid to be devoid of *Dam* methyltransferase targets (GATC), and we only replicated plasmids in a *Dcm* methyltransferase-deficient bacterial strain (GT115, Invivogen).

## Generation of plasmid libraries for mSTARR-seq

As input for our experiments with *pmSTARRseq1*, we isolated genomic DNA from the GM12878 cell line (QIAGEN, Blood and Cell Culture DNA Mini Kit) and fragmented the genomic DNA via sonication (Bioruptor Standard, Diagenode) or *MspI* digest. We size-selected both the sheared and *MspI*-digested DNA to a size range of ~300–700 bp, and used the NEBNext DNA Library Prep Master Mix Set for Illumina kit to create libraries from 1 to 2 ug of either sheared or *MspI*-digested DNA (completing 3–4 replicate library preps for input type). We followed the manufacturer's instructions except for the final PCR amplification step, during which we used the following primers to facilitate directional high-throughput cloning (F: actaaagtctagagcttgtaACACTCTTTCCCTACACG, R: gaagtggctggctgaattccGTGACTGGAGTTCAGACG).

We cloned each library (3–4 replicates per input type) into the *pmSTARRseq1* backbone using Gibson assembly. Specifically, we linearized the mSTARR-seq vector by digesting with *SpeI* and *NcoI* and isolating the backbone fragment (3.875 kb). We then assembled the *pmSTARRseq1* vector (1 ug) to an aliquot of *MspI*-digested or sheared library (284 ng) using 50 ul of NEBuilder HiFi DNA Assembly Master Mix in 100 ul reaction volumes. We completed 4 and 3 large-scale assemblies using *MspI*-digested and sheared DNA libraries, respectively. We incubated each of the 7 reactions for 60 min at 50°C, and purified the resulting reaction using a 1X Agencourt AMPure XP bead (Beckman Coulter) cleanup and an elution volume of 7.5 ul.

Each 7.5 ul aliquot of purified, assembled product (n = 7 large scale assemblies) was transformed into 300 ul of electrocompetent GT115 *E. coli* cells. GT115 cells were initially purchased from Invivogen, and a custom electrocompetent version of the strain was prepared by Intact Genomics (standard *E. coli* strains cannot be used in this protocol due to the R6Kγ origin of replication). We used the BioRad Gene Pulser Xcell system for electroporation and the *E. coli* 2 mm gap three kv program. Following electroporation, each of the 7 pools of 300 ul of cells was allowed to recover in 10 mL of SOC for 1 hr at 37°C. Each pool was then transferred to a flask containing 300 ml LB with Zeocin (100 ug/mL) and grown overnight for 12 hr. Plasmids were then extracted with the QIAGEN Plasmid Plus Maxi Kit. We pooled plasmid DNA derived from *MspI*-digested library transformations (n = 4) and sheared library transformations (n = 3), and mixed these two pools in a 1:2 ratio.

This 1:2 mixture (a total of 720 ug of plasmid DNA) was used to perform 6 'methylated condition' methyltransferase reactions (37°C overnight incubation of 60 ug of plasmid DNA with 50 ul of NEB buffer 2, 5 ul of 32 uM SAM, 9 ul of 20 U/ul *M.SssI* [New England Biolabs] in a 500 ul volume) and 6 'unmethylated condition' reactions in which the *M.SssI* enzyme was replaced with water. Following incubation, each replicate was purified using a 0.8X Agencourt AMPure XP bead cleanup. These 12 purified reactions were used as transfection input (see below) for the primary data set described in the main text.

For the additional set of experiments in which K562 cells were challenged or mock treated with IFNA, we used the same pool of post-transformation, extracted plasmid DNA (consisting of a mixture of *Msp1*-digested and sheared genomic DNA fragment inserts) described above. Using this pool, we performed an additional set of 2 'unmethylated condition' and 6 'methylated condition' methyltransferase reactions using the incubation and purification conditions described above.

## Cell culture, plasmid transfection, and cell harvesting

K562 cells were cultured in RPMI 1640 media (Gibco) supplemented with 10% FBS (Sigma Aldrich) and 1% Penicillin/Streptomycin solution (Sigma Aldrich) and maintained at 37°C in a 5% $CO_2$ incubator. All transfections were performed with Lipofectamine 3000 (ThermoFisher Scientific) following the manufacturer's instructions, with reagent quantities scaled as follows per replicate: 20 million cells, 40 ug of DNA, 110 ul of Lipofectamine 3000, and 100 ul of P3000. We transfected six unmethylated condition and six methylated condition replicates for the primary data set on mapping MD regulatory activity. For the additional experiments focusing on the IFNA response, we transfected an additional 2 and 6 replicates with unmethylated and methylated DNA, respectively, and added 2000 U/mL IFNA (ThermoFisher Scientific) to 3 of the methylated replicates 38 hr post-transfection. At the same time, we added an equivalent volume of water as a negative control to the remaining methylated and unmethylated replicates (total number of replicates: n = 3 methylated, IFNA+, n = 3 methylated, IFNA-, n = 2 unmethylated, IFNA-).

Total RNA and plasmid DNA was isolated from each K562 replicate cell population 48 hr post-transfection. Prior to cell lysis, each replicate was collected via centrifugation. Pelleted cells were washed with 1X PBS and then incubated at 37°C for 5–10 min in 3 mL RPMI 1640 containing 1 mL of Turbo DNase (Ambion) per 36 mL. Cells were then pelleted and washed with 1X PBS, and an aliquot of ~10% unlysed cells per replicate was set aside for later plasmid extraction. The remaining cells were pelleted and lysed in 2 mL of Buffer RLT (QIAGEN).

## Isolation and preparation of plasmid-derived mRNA and plasmid DNA libraries

Total RNA was extracted from each replicate (n = 12 from the main experiment, n = 8 from the IFNA experiment) using the QIAGEN RNeasy Midi kit, and the polyA +RNA fraction was isolated from up to 75 ug of total RNA using Dynabeads Oligo dT25 (Invitrogen) and eluted in 25 ul of 10 mM Tris-HCl. Each isolated mRNA sample was then DNase-treated by adding 1 ul of Turbo DNase, 3 ul of Turbo DNase buffer, and 1 ul of water, followed by a 30-min incubation at 37°C and purification using the RNeasy MiniElute Cleanup Kit (QIAGEN). Purified mRNA was eluted in 25 ul of RNase free water, after which reverse transcription was performed using Super Script III Reverse Transcriptase (Invitrogen) following the manufacturer's recommended protocol for higher yield. First strand cDNA synthesis was performed with a primer specific to the mSTARR-seq reporter plasmid (CAAAC TCATCAATGTATCTTATCATG), including the optional RNase H step. cDNA from each replicate was purified and concentrated using a 2X Agencourt AMPure XP bead cleanup. Each sample was eluted in 20 ul of nuclease free water.

We amplified the cDNA obtained from reverse transcription for Illumina sequencing using a two step, nested PCR strategy. In the first PCR, we amplified each cDNA pool (n = 12 from the main experiment, n = 8 from the IFNA experiment) using two primers specific to the mSTARR-seq reporter plasmid (forward: GGGCCAGCTGTTGGGGTG*T*C*C*A*C (3' end is protected by phophorothioate bonds), reverse: CAAACTCATCAATGTATCTTATCATG). The forward primer spans the splice junction of the synthetic intron and specifically amplifies the reporter cDNA without amplifying any residual, unspliced plasmid DNA left in the reaction mixture. Each PCR reaction contained 2.5 ul of the forward and reverse primers at 10 uM, 25 ul of NEBNext High-Fidelity 2X PCR Master Mix, and 20 ul of cDNA. Cycling conditions were as follows: 98°C for 2 min; followed by 15 cycles of 98°C for 20 s, 65°C for 30 s, 72°C for 120 s; followed by a final 5 min extension at 72°C for 5 min. PCR products were purified using a 2X Agencourt AMPure XP bead cleanup and eluted in 20 ul of EB buffer (QIAGEN).

The entire purified PCR product (20 ul) from each reaction (n = 12 from the main experiment, n = 8 from the IFNA experiment) served as the template for the second PCR, which contained 25 ul of NEBNext High-Fidelity 2X PCR Master Mix, 2.5 ul of universal primer, and 2.5 ul of an indexed primer (NEBNext Multiplex Oligos for Illumina). Cycling conditions were as described above (but using 10 cycles instead of 15). PCR products were purified using a 1.2X Agencourt AMPure XP bead cleanup.

To measure the relative abundance of each DNA fragment in the input plasmid libraries, we created DNA-seq libraries from plasmid DNA extracted from each replicate 48 hr post transfection. 10 ng of plasmid DNA was used as the input for 50 ul PCRs each containing 25 ul of NEBNext High-Fidelity 2X PCR Master Mix, 2.5 ul of universal primer, and 2.5 ul of an indexed primer (NEBNext Multiplex Oligos for Illumina). Cycling conditions were as described above (using 10 cycles) and PCR products were purified using a 1X Agencourt AMPure XP bead cleanup. All final DNA-seq and mRNA-seq libraries were quantified on an Agilent DNA High Sensitivity Chip and sequenced on the Illumina 4000 platform using 100 bp PE reads (see *Supplementary file 1* for sample-specific read depths).

## Generation and low-level processing of bisulfite sequencing libraries

To confirm the DNA methylation status of plasmids in the main experiment (testing for MD activity across six unmethylated and six methylated condition replicates), we used 200 ng of plasmid DNA extracted from each K562 replicate cell population as the input for multiplexed bisulfite sequencing (*Meissner et al., 2008*; *Gu et al., 2011*; *Boyle et al., 2012*). To do so, we first fragmented plasmid DNA by overnight incubation (at 37°C) with 1 ul *BfaI*, 1 ul *HindIII*, 1 ul *HpaI*, and 3 ul Cutsmart buffer

(New England BioLabs) in a total reaction volume of 30 ul. Approximately 1 ng of unmethylated phage DNA (Sigma Aldrich) was spiked in with each sample for downstream assessment of the bisulfite conversion efficiency. We performed end repair, A-tailing, and adapter ligation following previously published protocols (*Boyle et al., 2012*). After adapter ligation, all samples (n = 12) were pooled together and subjected to two rounds of sodium bisulfite conversion using the EpiTect Bisulfite Conversion kit (QIAGEN). Libraries were then PCR amplified and purified as in *Boyle et al. (2012)*, quantified on a Agilent DNA High Sensitivity Chip (Agilent Bioanalyzer 2100), and sequenced on the Illumina MiSeq platform (see *Supplementary file 1* for sample-specific read depths).

We processed each bisulfite sequencing library as follows. First, we removed adapter contamination, low-quality bases, and bases artificially introduced during library construction using the program *cutadapt*. We then isolated CpG sites introduced through Gibson assembly (the Illumina-style adapters used for cloning and sequencing contain CpG sites) and estimated CpG methylation levels directly from these introduced sequences. In addition, we used BSMAP (*Xi and Li, 2009*) to map reads to the lambda phage genome. Following mapping, we extracted the methylated read count and total read count for each plasmid library and CpG site using a Python script packaged with BSMAP (*Xi and Li, 2009*). We then estimated the conversion efficiency of each bisulfite sequencing library (*Becker et al., 2011*; *Banovich et al., 2014*; *Lea et al., 2016*); all conversion efficiency estimates exceeded 99.7% (*Supplementary file 1*).

For plasmid DNA extracted from the IFNA experimental replicates (n = 3 methylated, IFNA+, n = 3 methylated, IFNA-, n = 2 unmethylated, IFNA-), we aimed to generate higher coverage bisulfite-sequencing libraries to facilitate locus-specific analysis. To do so, we digested the plasmid DNA extracted from each replicate with *BfaI* using the conditions described above. We ran the digestion products on a 2% agarose gel and extracted the 250–650 bp region containing human DNA inserts as well as digested plasmid backbone (QIAquick Gel Extraction Kit, QIAGEN). Two of the eight replicates (n = 1 methylated, IFNA +replicate and one methylated, IFNA- replicate) did not yield detectable amounts of gel-extracted, size-selected DNA and were dropped from further library preparation. The remaining samples (ranging from 25 to 46 ng total), were each spiked with a small amount of unmethylated phage DNA, and this mixture was used as input for bisulfite-sequencing library preparation. To accommodate the low input, we used the NEXTflex Bisulfite-Seq Kit (Bioo Scientific) according to the manufacturer's instructions. Sequencing output (from a HiSeq 4000 instead of a MiSeq, due to the need for increased coverage) from these six libraries were processed as described above with the following exceptions: (i) we only analyzed reads that showed evidence of proper *BfaI* digestion and that started with the plasmid sequence we expected to precede the cloning site for human insert fragments; (ii) we trimmed the plasmid sequence from the human insert fragments prior to mapping; and (iii) we mapped to a combined reference that included the human genome (*hg38*), the *pmSTARRseq1* plasmid sequence, and the unmethylated phage genome.

## Low-level data processing of mRNA-seq and DNA-seq libraries

Following sequencing, we removed adapter contamination and low-quality bases from each library using *cutadapt*. We mapped the trimmed reads to the human reference genome (*hg38*) using BWA (*Li and Durbin, 2009*), and retained only uniquely mapped reads. We estimated the number of unique fragments assayed in our experiments by counting the number of RNA fragments with unique start and end positions within each replicate (*Supplementary file 1*; note that we excluded mRNA and DNA-seq data for one replicate from all analyses because of low sequencing depth).

For all analyses, we focused on counts of the number of mapped reads in each library that overlapped 200 bp non-overlapping genomic intervals created with the BEDtools function 'makewindows' (*Quinlan and Hall, 2010*). Pileups were determined with the 'coverage' function. For the main experiment (i.e. excluding IFNA experiment samples; n = 11 DNA-seq and 11 mRNA-seq libraries), we removed 200 bp regions that had zero mRNA counts in three or more replicates per condition, as well as regions that had low stability and repeatability of DNA counts. Specifically, to identify high variability regions, we first normalized the filtered DNA library count matrix using the function 'voomWithQualityWeights' from the R package 'limma' (*Law et al., 2014*). We then computed the distribution of pairwise differences in abundance between all pairs of DNA replicates. Next, we removed regions from our analysis if >25% of pairs fell outside the central 90th percentile of this distribution. This filtering approach resulted in 262,829 200 bp regions.

## Identification of enhancers and methylation dependent (MD) enhancers

Using the set of 262,829 200 bp analyzable regions from our main dataset, we first identified regions that exhibited self-transcribing regulatory activity (more mRNA relative to DNA input). Specifically, for each 200 bp region, we ran a nested model on the normalized values that estimated differences between mRNA and input DNA abundance within each condition:

$$y_i = \mu + t_i\beta_1 * I(c_i = 0) + t_i\beta_2 * I(c_i = 1) + \varepsilon_i \tag{1}$$

where $y_i$ is the normalized count value for sample $i$, $t_i$ is the sample type (input DNA or mRNA), and $I$ is an indicator variable for condition (where $c_i = 0$ or $c_i = 1$ indicates a sample from the unmethylated or methylated condition, respectively). $\beta_1$ and $\beta_2$ are thus estimates of the strength of the differences between mRNA and input DNA abundance in the unmethylated and methylated conditions, respectively, where beta values > 0 indicate that mRNA abundance is greater than input DNA abundance. $\mu$ is the model intercept and $\varepsilon_i$ denotes model error. We extracted the p-values associated with $\beta_1$ and $\beta_2$ from each model, and corrected for multiple hypothesis testing using the R function 'qvalue' (*Dabney and Storey, 2015*). We considered a region to have enhancer activity if $\beta_1$, $\beta_2$, or both were greater than 0 and had an FDR-corrected p-value less than 0.1 (*Supplementary file 3*).

Using these criteria, we identified 19,703 (7.5%) regions with enhancer activity. We next tested for methylation dependent (MD) activity within this set of enhancers by adding an interaction term between condition (unmethylated or methylated) and sample type (input DNA or mRNA):

$$y_i = \mu + t_i\beta_t + c_i\beta_c + (t_i \text{ x } c_i)\beta_{txc} + \varepsilon_i \tag{2}$$

where $y_i$, $t_i$, $c_i$ and $\varepsilon_i$ are defined as in equation 1. $\beta_t$ is the effect of sample type (input DNA or mRNA), $\beta_c$ is the effect of condition (unmethylated or methylated), and $\beta_{txc}$ is the estimate of the interaction between the two variables ($t_i$ x $c_i$). We extracted the p-value associated with $\beta_{txc}$ from each model, and corrected for multiple hypothesis testing using the R function 'qvalue' (*Dabney and Storey, 2015*). We considered a region to have methylation-dependent enhancer activity if the estimate for $\beta_{txc}$ passed an FDR threshold of 10% (*Supplementary file 4*; see also Q-Q plot in *Figure 3—figure supplement 1*).

## Testing for type I interferon response

Recent work (*Muerdter et al., 2018*) has shown that reporter assays that rely on plasmid transfection into human cells may be affected by two technical sources of error: (i) the bacterial plasmid origin-of-replication (ORI) can serve as a conflicting promoter that competes with the plasmid's eukaryotic promoter and (ii) transfection of plasmid DNA into the cell may activate its type I interferon (IFN-I) response. Problem (i) is unlikely to be an issue for mSTARR-seq because the *pmSTARR-seq1* plasmid contains the R6Kγ origin of replication, unlike typical plasmids used for high-throughput reporter assays (e.g. STARR-seq (*Arnold et al., 2013*) and MPRAs (*Melnikov et al., 2012*)), which are built from the popular pGL3/4 family of plasmids. Importantly, when we applied a computational algorithm for prediction of eukaryotic PolII promoters (*Knudsen, 1999*) to pGL3 and *pmSTARRseq1*, a transcription start site was predicted within the pGL3 origin of replication but not within the *pmSTARRseq1* origin of replication.

Problem (ii) is also unlikely to affect our mSTARR-seq experiments because K562s have inactive cGAS/STING pathways (the major avenue through which cells sense cytoplasmic DNA and induce an IFN-I response (*Muerdter et al., 2018*)). To test this hypothesis, we performed qPCR for 6 IFN-I response genes and two control genes on total RNA extracted from K562s that either (i) had not been transfected with plasmid, or (ii) had been transfected with an unmethylated or methylated *pmSTARRseq1* pool. We performed qPCR on RNA from three replicate cell populations and RNA extractions for each condition (untransfected, transfected with methylated DNA, or transfected with unmethylated DNA), using the primer sequences and methods from *Muerdter et al. (2018)*.

In agreement with previous work, we found little evidence that plasmid transfection induced an IFN-I response in the K562 cell line (two-sided Wilcoxon signed-rank test comparing RNA abundance in untransfected versus transfected cells: p>0.05 for all genes tested except *OAS3*, where p=0.047; *Figure 1—figure supplement 4*). In addition, we used Gene Ontology (GO) annotations and the

program PANTHER (*Mi et al., 2016*) to ask whether regions with significant mSTARR-seq activity in the unmethylated condition were more likely to fall near (within 100 kb of the transcription start or end site) genes involved in the IFN-I response or other immune-related pathways, compared to the background set of all genes near a region analyzed in our K562 data set. We also asked whether the strongest unmethylated condition enhancers (top decile) were more likely to fall near genes involved in particular GO categories compared to the background set of all genes near significant mSTARR-seq enhancers. No GO categories associated with the IFN-I response or immune processes were observed.

## Luciferase reporter assays

We chose 18 candidate regions that spanned a range of mSTARR-seq activity patterns (*Supplementary file 5A*). For each region, we synthesized the human *hg38* DNA sequence flanked by 15 base pairs complementary to the 5' and 3' ends of the pCpGL backbone (following linearization of the pCpGL plasmid with *PstI* [New England BioLabs] and purification using a 1.5X Agencourt AMPure XP bead cleanup [Beckman Coulter]). We ligated each synthesized fragment to the linearized pCpGL backbone using Gibson assembly (Gibson Assembly Master Mix, New England BioLabs). Each ligation reaction was purified using a 1.5X Agencourt AMPure XP bead cleanup (Beckman Coulter), and the resulting reactions were chemically transformed into competent *E. coli* GT115 cells (InvivoGen), selected and grown on LB agar plates in the presence of Zeocin, and purified using the QIAprep Spin Miniprep Kit (QIAGEN).

We subjected the purified plasmid DNA (containing the pCpGL backbone and a human sequence insert) to two treatments: (i) in vitro methylation by *M.SssI* methyltransferase in the presence of the methyl donor S-adenosylmethionine (SAM), which results in methylation of all CpGs; or (ii) a mock treatment using only SAM and no methyltransferase. We confirmed that the human sequence insert was methylated as expected by digesting each treated plasmid with *MspI*, a methylation insensitive enzyme that targets the sequence motif CCGG, and *HpaII*, an isoschizomer of *MspI* whose activity is blocked by DNA methylation.

We transfected each methylated or unmethylated plasmid construct into the K562 line. Specifically, when cells were approximately 70% confluent, we washed and seeded them in pools of 25,000 cells. Transient transfection was performed by adding 20 uL of OPTIMEM Reduced Serum Media (Gibco) containing the following reagents: (i) 100 ng of methylated or unmethylated vector (eight replicates for each condition); (ii) 10 ng of Renilla control vector; (iii) 0.5 uL of Lipofectamine; and (iv) 0.1 uL of the PLUS reagent (from the Lipofectamine 2000 system, Life Technologies). Cells were incubated for 72 hr following transfection, and subsequently assayed for transgene luciferase expression with the Dual Luciferase Assay kit (Promega; full results presented in *Supplementary file 5B*). We excluded 1–2 replicates for certain constructs because of obvious liquid evaporation during luciferase expression measurement (resulting in a final data set with 6–8 replicates per construct).

We quantified enhancer activity by first normalizing firefly luciferase activity against renilla luciferase activity to control for variation in transfection efficiency or cell number. We then asked whether human DNA fragments with strong enhancer activity in the unmethylated condition in mSTARR-seq also displayed strong enhancer activity in our validation experiments (specifically, in the 'sham' condition where the sequence was unmethylated). To do so, we used linear mixed effects models to predict $\log_2$ normalized luciferase activity as a function of the synthesized fragment length (because larger query fragments tend to have stronger activity), the estimate of enhancer strength in the unmethylated condition from mSTARR-seq (provided in *Supplementary file 3*), and a random effect of assay batch. To ask whether fragments with MD enhancer activity in mSTARR-seq also displayed MD enhancer activity in our validation experiments, we used a similar modeling approach with $\log_2$ normalized luciferase activity in the unmethylated condition divided by $\log_2$ normalized luciferase activity in the methylated condition as the outcome variable. In this case, our model included the synthesized fragment length, the estimate of MD activity from mSTARR-seq (provided in *Supplementary file 4*), and a random effect of assay batch.

## Annotation of analyzed mSTARR-seq fragments

Each 200 bp region was originally assayed on one or more ~300 bp fragments. Thus, focusing our annotations on the 200 bp analyzed window alone could miss adjacent genomic features that are

responsible for observed enhancer or MD enhancer activity. Therefore, for all annotations, we focused on the window between the start and end position of each assayed DNA fragment that contained a given 200 bp region. In cases where multiple overlapping DNA fragments contained a given 200 bp region, we used the genomic coordinates of the most downstream and most upstream start and end position, respectively.

Using these start and end coordinates for each 200 bp region covered in our experiments, we calculated the proportion of each of 12 K562-annotated ENCODE (*Karolchik et al., 2014*) ChromHMM classes overlapping the fragments we analyzed using the BEDtools (*Quinlan and Hall, 2010*) function 'intersect'. ChromHMM annotations were downloaded from http://hgdownload.cse.ucsc.edu/goldenPath/hg19/encodeDCC/wgEncodeBroadHmm/. We assigned the fragment window associated with each 200 bp region to a particular chromatin state if >50% of the window overlapped that particular state and performed two-sided Fisher's exact tests for each chromatin state (see *Figure 1—figure supplement 4* for results of these analyses).

To perform our random forests prediction, we gathered the following data for each fragment window associated with each analyzed 200 bp region:

1. The expression level of the nearest gene for regions that were within 200 kb of a gene TSS or TES. Expression levels were summarized as FPKM estimates from RNA-seq performed on K562 cells (NCBI GEO accession GSE86747).
2. The average CpG methylation level of the fragment window in K562s. Estimates were derived from ENCODE whole genome bisulfite sequencing data, after removing low coverage (<5 x) CpG sites (NCBI GEO accession GSM958729).
3. The average DNase-seq signal for the fragment window in K562s, using DNase-seq peak strength (downloaded from http://hgdownload.cse.ucsc.edu/goldenPath/hg38/database/wgEncodeRegDnaseUwK562Peak.txt.gz).
4. The average evolutionary conservation score of the region, using phastCons scores derived from a 44-way alignment across placental mammals (http://hgdownload.cse.ucsc.edu/goldenPath/hg19/database/phastConsElements46wayPlacental.txt.gz).
5. The absolute number of CpG sites and CpG density in the fragment window based on the human reference genome (hg38) sequence.
6. The presence or absence of a transcription factor binding site as determined by 127 unique experiments. For these annotations, we relied on publicly available ENCODE ChIP-seq data generated in K562s (downloaded from http://hgdownload.soe.ucsc.edu/goldenPath/hg19/encodeDCC/wgEncodeAwgTfbsUniform/). We downloaded all available ChIP-seq experiments (including replicate experiments for the same TF conducted by different labs), but did not include ChIP-seq data derived from stimulated/treated cells.

## Random forests classification

To determine whether MD activity could be reliably predicted from a set of sequence and functional genomic feature annotations for each 200 bp region (*Supplementary file 6*), we conducted random forest analyses using our main data set on detecting MD regulatory elements. We compared MD enhancers with greater activity when unmethylated against enhancers with no evidence for methylation dependence in the K562 cell line (FDR > 50% in the test for MD activity). Because random forests require a complete data set, we removed regions with missing values for any of the predictive features, resulting in a data set of n = 2317 MD enhancers and n = 2657 non-MD enhancers. We did not attempt a parallel analysis using MD enhancers with greater activity when methylated because of the relatively small data set of sequences that exhibited this pattern.

We then used the R package 'randomForest' (*Liaw and Wiener, 2002*) to iteratively construct training sets comprised of approximately 2/3 of the original data, and test sets comprised of the remaining enhancers in the data set (the 'out of bag' set). We grew 1000 classification trees and evaluated predictive accuracy using the out of bag test sets (note that although the set of regions used as the training versus test set differ in each iteration, for any given run, the test set were completely held out during model training). We then assigned a given region to a given predicted outcome class if the majority (>50%) of trees 'voted' for this class (see *Figure 3—figure supplement 2* for true positive/false positive trade-offs using alternative thresholds). To estimate the predictive value of individual features, we calculated the mean decrease in accuracy and the Gini coefficient, and estimated the significance of these contributions via comparison to the results of permutation

analyses (1000 permutations of MD versus non-MD classification) implemented in the R package 'rfPermute' (*Archer, 2015*). We considered a variable to be significant if the mean decrease in accuracy and Gini coefficient estimates both passed a 10% empirical FDR (see *Figure 3* for significant variables; see *Supplementary file 6* for information on all features). We note that these importance measures do not account for correlations between features (e.g., CpG number, CpG density, and methylation level). None of our variables are perfectly collinear, however, and our results in *Figure 3* indicate that all significantly predictive features contain some independent information (otherwise the accuracy of the RF would not drop when one feature was removed).

## Transcription factor binding motif enrichment analyses

To ask whether binding sites for certain TFs were consistently associated with windows with MD enhancer activity, we used the 'findMotifsGenome.pl' script in the motif analysis program HOMER (*Heinz et al., 2010*) to test for enrichment of 364 known vertebrate binding motifs relative to the background set of fragment windows associated with all mSTARR-seq enhancers. We performed this analysis twice, focusing on test sets of fragment windows associated with MD enhancers that were more active when unmethylated and more active when methylated, respectively. We considered a TF-binding motif to be significantly enriched in each of the test sets if the motif was found on >10% of fragments in the background set and the motif passed a 1% FDR threshold (*Benjamini and Hochberg, 1995*). Results from our motif enrichment analyses are provided in *Figure 4* and *Supplementary file 7–8*. To test for enrichment of specific TF families among the set of significant motifs identified in the K562 data set, we used hypergeometric tests to compare the proportion of significant motifs belonging to a particular TF family (e.g. ETS, GATA) to the proportion of all tested motifs that belong to that family.

## Analysis of locus-specific DNA methylation data

We generated bisulfite sequencing libraries across six replicates from our IFNA experiments (n = 2 unmethylated, IFNA-; n = 2 methylated, IFNA+; n = 2 methylated, IFNA-). Using data from the IFNA- samples, we estimated mean methylation levels for the plasmid backbone (where we expect no methylation) and human insert fragments separately. To ask whether the endogenous methylation level of a given CpG site affects the likelihood of that site gaining or losing DNA methylation marks post-transfection, we used ENCODE whole genome bisulfite sequencing data for K562s (NCBI GEO accession GSM958729). Specifically, for IFNA- unmethylated and methylated replicates separately, we calculated the Spearman correlation between the mean post-transfection methylation level of each CpG site sequenced at $\geq 1$ x coverage in at least one mSTARR sample and its endogenous methylation level in K562s (for CpG sites sequenced at >5 x coverage in the ENCODE dataset; n = 201,524 and 91,103 sites analyzed for the unmethylated and methylated replicates, respectively). To understand whether CpGs in certain TFBS were more likely to gain or lose methylation post-transfection, we estimated the mean methylation level of all CpGs in binding sites identified from K562 ChIP-seq data, for 127 TFs separately (see item (v) in *Annotation of analyzed mSTARR-seq fragments*). We excluded TFs from this analysis if we did not cover at least 100 CpGs in binding sites for the focal TF, leaving us with 123 and 121 analyzable TFs for unmethylated and methylated samples, respectively. Finally, we used hypergeometric tests to investigate enrichment of individual TF families among the top 20 TFs associated with the greatest change in DNA methylation post-transfection (increase in methylation in the unmethylated condition or demethylation in the methylated condition), using TF family annotations from HOMER (*Heinz et al., 2010*).

## Correlations between DNA methylation and gene expression levels in primary cells and after IFNA treatment

To understand whether mSTARR-seq data explains heterogeneity in in vivo DNA methylation-gene expression correlations across individuals, in primary cells, we used a publicly available data set on 1202 monocyte samples (*Reynolds et al., 2014*). We downloaded paired genome-wide DNA methylation data (measured on the Illumina HumanMethylation450 BeadChip) and gene expression data (measured on the Illumina HumanHT-12 v4 Expression BeadChip) for each sample. DNA methylation data were provided as continuously varying, unbounded M-values (*Du et al., 2010*) and expression

data were provided as raw signal intensity values, which we normalized using the *voom* function in the *limma* R package (*Law et al., 2014*)).

We filtered the DNA methylation data and expression data to remove probes with mean detection level p-values>0.05. We further filtered the DNA methylation data to remove probes with M-values corresponding to an average methylation level <0.1 or>0.9, which represent constitutively hypo- and hypermethylated regions, respectively. This filtering left us with 15,247 probes measuring gene expression and 425,895 probes measuring DNA methylation. We next identified CpG sites that overlapped regions with significant enhancer activity in our mSTARR-seq assay. We associated each of these CpG sites to its closest gene and removed CpG sites from our analysis that were >100 kb from any gene. In the remaining set of 2699 unique CpG sites (associated with 1002 unique genes), we used linear models to test for an association between methylation levels at each filtered CpG site and gene expression levels. We extracted the p-value and estimated coefficient from each model, and considered a correlation to be significant if it passed a 10% FDR. Finally, we annotated each CpG based on whether it overlapped an MD or non-MD enhancer. We used two-sided Fisher's Exact Tests to investigate enrichment of CpG methylation-expression correlations in MD versus non-MD mSTARR-seq enhancers.

Finally, to confirm that DNA methylation-gene expression correlations arise following major environmental perturbations to gene expression, we generated mRNA-seq, DNA-seq, and BS-seq data from K562s transfected with methylated libraries and treated with either IFNA (IFNA+) or water (the control condition: IFNA-). After confirming there were no systematic differences in plasmid DNA input between replicates (all $R^2$ >0.98, linear models comparing limma-normalized counts between all pairs of replicates), we used the mRNA-seq data to estimate IFNA effects on enhancer activity. Specifically, we calculated the mean difference in limma-normalized counts between IFNA + and IFNA - samples for each 200 bp region captured in at least one replicate per condition (n = 217,439 regions). To estimate IFNA effects on plasmid DNA methylation levels, we calculated the mean difference in post-transfection methylation levels between IFNA + and IFNA - samples, for CpGs covered by at least one replicate per condition and that also overlapped the set of 217,439 analyzable 200 bp regions (n = 7010 CpG sites from 2678 200 bp regions). We then calculated the Spearman correlation between the effect of the IFNA challenge on enhancer activity and the effect of IFNA on mean plasmid methylation levels, for all 200 bp regions and for 200 bp regions that showed progressively more strongly induced enhancer activity (*Figure 5—figure supplement 1*). Finally, we used logistic regression to test whether IFNA-induced changes in enhancer activity predicted the probability of demethylation after IFNA treatment, where sites were categorized into those that either retained (0) or lost (1) their initial, *M.SssI*-induced methylation status.

## Cell lines

All experiments described here used the K562 cell line (ATCC CCL-243). Cells were obtained from Duke University's Cell Culture Facility and were tested by the facility for mycoplasm contamination and cell line identity (through STR profiling).

## Data availability

Accession numbers and links for publicly available data sets used for analyses are provided in the Materials and methods. All sequencing data generated as part of this work are available through NCBI's Short Read Archive (SRP120556). The mSTARR-seq protocol is available online at www.tung-lab.org/protocols-and-software.html. The DNA input library described here is available on request from the authors, and the *pmSTARRseq1* vector is available through AddGene.

## Acknowledgements

We thank Michael Yuan, Tawni Voyles, Graham Johnson, Alain Pacis, Vania Yotova, and members of the Reddy and Tung labs for experimental contributions and helpful discussions; the Rehli lab for the gift of the pCpGL vector; and three anonymous reviewers for constructive comments on a previous version of this manuscript. This work was supported by a Sloan Foundation Early Career Research Fellowship and a Foerster-Bernstein postdoctoral fellowship; NIH grants R01-GM102562, R01-HD088558, R21-AG049936, U01-HG007900, UM1-HG009428, and F31-HL129743; NSF grant BCS-1455808; and North Carolina Biotechnology Center grant 2016-IDG-1013.

## Additional information

### Funding

| Funder | Author |
| --- | --- |
| Alfred P. Sloan Foundation | Jenny Tung |
| National Institutes of Health | Luis B Barreiro<br>Timothy E Reddy<br>Jenny Tung |
| National Science Foundation | Amanda Lea<br>Jenny Tung |

The funders had no role in study design, data collection and interpretation, or the decision to submit the work for publication.

### Author contributions

Amanda J Lea, Conceptualization, Data curation, Formal analysis, Supervision, Investigation, Visualization, Methodology, Writing—original draft, Writing—review and editing; Christopher M Vockley, Conceptualization, Resources, Supervision, Investigation, Methodology, Writing—review and editing; Rachel A Johnston, Christina A Del Carpio, Investigation, Writing—review and editing; Luis B Barreiro, Conceptualization, Supervision, Writing—review and editing, Funding acquisition; Timothy E Reddy, Conceptualization, Resources, Supervision, Writing—review and editing, Funding acquisition; Jenny Tung, Conceptualization, Supervision, Funding acquisition, Writing—original draft, Writing—review and editing

### Author ORCIDs

Amanda J Lea  http://orcid.org/0000-0002-8827-2750
Timothy E Reddy  https://orcid.org/0000-0002-7629-061X
Jenny Tung  http://orcid.org/0000-0003-0416-2958

### Decision letter and Author response

Decision letter https://doi.org/10.7554/eLife.37513.045
Author response https://doi.org/10.7554/eLife.37513.046

## Additional files

### Supplementary files

• Supplementary file 1. Table describing the samples sequenced as part of this study (contains source data for *Figure 1E*).
DOI: https://doi.org/10.7554/eLife.37513.020

• Supplementary file 2. Table showing that plasmid transfection does not induce a strong type I interferon responses in the K562 cell line (by testing GO enrichments of the strongest mSTARR-seq identified enhancers).
DOI: https://doi.org/10.7554/eLife.37513.021

• Supplementary file 3. Table of results from nested model testing for enhancer activity.
DOI: https://doi.org/10.7554/eLife.37513.022

• Supplementary file 4. Table of results from interaction model testing for MD enhancer activity (contains source data for *Figure 3A*).
DOI: https://doi.org/10.7554/eLife.37513.023

• Supplementary file 5. Table describing. (A) luciferase reporter assay construct information and (B) luciferase reporter assay results (contains source data for *Figure 2C*).
DOI: https://doi.org/10.7554/eLife.37513.024

• Supplementary file 6. Table of random forests analysis results (contains source data for *Figure 3F*).
DOI: https://doi.org/10.7554/eLife.37513.025

- Supplementary file 7. Table of TF motif enrichment results for MD enhancers with greater activity in the unmethylated condition in the K562 cell line (contains source data for *Figure 4A*).
DOI: https://doi.org/10.7554/eLife.37513.026

- Supplementary file 8. Table of TF motif enrichment results for MD enhancers with greater activity in the methylated condition in the K562 cell line (contains source data for *Figure 4B*).
DOI: https://doi.org/10.7554/eLife.37513.027

- Supplementary file 9. Table of mean methylation levels for CpG sites in ChIP-seq identified TF binding sites.
DOI: https://doi.org/10.7554/eLife.37513.028

- Transparent reporting form
DOI: https://doi.org/10.7554/eLife.37513.029

### Data availability

Accession numbers and links for publicly available data sets used for analyses are provided in the Methods. All sequencing data generated as part of this work are available through NCBI's Short Read Archive (SRP120556). The mSTARR-seq protocol is available online at www.tung-lab.org/proto-cols-and-software.html. The plasmid DNA input library described here is available on request from the authors (sequence data for the same plasmid DNA input library is available through the SRA deposit), and the pmSTARRseq1 vector is available through AddGene.

The following dataset was generated:

| Author(s) | Year | Dataset title | Dataset URL | Database and Identifier |
|---|---|---|---|---|
| Amanda Lea | 2018 | Genome-wide quantification of the effects of DNA methylation on human gene regulation | https://www.ncbi.nlm.nih.gov/sra/?term=SRP120556 | NCBI Sequence Read Archive, SRP120556 |

The following previously published datasets were used:

| Author(s) | Year | Dataset title | Dataset URL | Database and Identifier |
|---|---|---|---|---|
| ENCODE Project Consortium | 2012 | An integrated encyclopedia of DNA elements in the human genome, wgEncodeBroadHmmK562HMM.bed.gz | http://hgdownload.cse.ucsc.edu/goldenPath/hg19/encodeDCC/wgEncodeBroadHmm/ | hgdownload, hg19 |
| ENCODE Project Consortium | 2012 | An integrated encyclopedia of DNA elements in the human genome | https://www.encodeproject.org/experiments/ENCSR765JPC/ | ENCODE, ENCSR765JPC |
| ENCODE Project Consortium | 2012 | An integrated encyclopedia of DNA elements in the human genome | https://www.encodeproject.org/experiments/ENCSR765JPC/ | ENCODE, ENCSR765JPC |
| ENCODE Project Consortium | 2012 | An integrated encyclopedia of DNA elements in the human genome, wgEncodeRegDnaseUwK562Peak.txt | http://hgdownload.cse.ucsc.edu/goldenPath/hg38/database/wgEncodeRegDnaseUwK562Peak.txt.gz | hgdownload, hg38 |
| Donna Karolchik, Galt P. Barber, Jonathan Casper, Hiram Clawson, Melissa S. Cline, Mark Diekhans, Timothy R. Dreszer, Pauline A. Fujita, Luvina Guruvadoo, Maximilian Haeussler, Rachel A. Harte, Steve Heitner, Angie S. Hinrichs, Katrina Learned, Brian T. Lee, Chin H. Li, Brian J. Raney, | 2014 | The UCSC Genome Browser database: 2014 update, phastConsElements46wayPlacental.txt | http://hgdownload.cse.ucsc.edu/goldenPath/hg19/database/phastConsElements46wayPlacental.txt.gz | hgdownload, hg19 |

| | | | | | |
|---|---|---|---|---|---|
| Brooke Rhead, Kate R. Rosenbloom, Cricket A. Sloan, Matthew L. Speir, Ann S. Zweig, David Haussler, Robert M. Kuhn, W. James Kent | | | | | |
| ENCODE Project Consortium | 2012 | An integrated encyclopedia of DNA elements in the human genome, All K562 narrow peak files | http://hgdownload.soe.ucsc.edu/goldenPath/hg19/encodeDCC/wgEncodeAwgTfbsUniform/ | | hgdownload, encodeDCC |

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
