## [Decision Letter]

Thank you for sending your article entitled "Genome-wide quantification of the effects of DNA methylation on human gene regulation" for peer review at *eLife*. Your article is being evaluated by three peer reviewers and the evaluation is being overseen by a Reviewing Editor and Detlef Weigel as the Senior Editor.

Given the list of essential revisions requested, including potentially new experiments, the editors and reviewers invite you to respond within the next two weeks with an action plan and timetable for the completion of the additional work. We plan to share your responses with the reviewers and then issue a binding recommendation.

We request that your plan to revise the manuscript includes an analysis, using existing data, of whether the enhancers, bound by methylation-independent binders, are methylated at the endogenous loci (potentially using DNA methylation data from K562). If they are found to be endogenously de-methylated, then you will need to explain why the endogenous enhancer is hypomethylated, while the episomal constructs remain methylated, even when the same transcription factors are bound at both sites. If such data are shown, interpreted and discussed, then the reader can better understand this issue.

The reviewers noted that no strong correlation between methylation and expression was reported previously, yet differential methylation does correlate with gene expression variance. For example, an increase in gene expression is correlated with a decrease in methylation of promoter and/or enhancers that underlie this up-regulation. To take this into account, the reviewers requested a plan for further measurements in the reporter assay, along with the methylation state of the episomal constructs during a state transition, such as cell differentiation, or a treatment (e.g., a drug treatment of K562, which causes large transcriptional changes) in which changes in transcription factor binding and consequent effects are expected. We finally note that the prediction of motifs in methylation dependent vs. independent enhancers is good, but remains at the prediction level, and is not validated experimentally.

The reviewers also request a plan for extended analyses on assay coverage and other aspects of the computational analysis (see reviewer #3's comments).

Reviewer #1:

The authors propose a new experimental approach, mSTARR-Seq, for testing the causal effects of DNA methylation on gene expression at millions of methylation CpG sites. They find that few sites are 'methylation dependent' but those that are MD are predictable from chromatin state measurements. The approach, to my knowledge, is novel, robust and produces important new findings. I do not have substantive concerns. The comments that I provide below are thus not intended to imply a lack of support for this manuscript.

I was surprised that the 8% figure in Results paragraph three was not emphasised more. It is important, to my mind, that the reader is told explicitly about this 92%:8% split. Previous statements, such as "our observations indicate a substantial degree of coordination between methylation levels and gene expression" (Banovich et al.) might be interpreted as implying otherwise.

The RF approach predicted MD regulatory element activity with ~79% accuracy, with 26 features being significantly contributing. The MD transcription factor-DNA binding findings (summarised in Figure 4) are complementary to similar findings from others, e.g. Yin et al., 2017 and Table 1.

"We also show that MD regulatory elements explain site-to-site heterogeneity in the strength of DNA methylation-gene expression correlations measured within a population sample". Can the authors be clearer quantitatively how explanatory this is?

Reviewer #2:

Lea et al. describe mSTARR-seq: STARR-seq using a library of enhancers, whereby the library is either methylated (in bacteria) or unmethylated. Comparison of enhancer reporter levels between both conditions, allows identifying methylation-dependent enhancers, as well as methylation-independent. What is interesting about this study is that it aims to provide a direct measurement of the causality of DNA methylation of an enhancer/promoter sequence in driving gene expression. However, the interpretation of the data is not fully convincing, there is little follow-up, and because of that the study, in its current form, has limited depth. This is a difficult problem, so the attention and approach is timely, but given the confusion and contradictions in the field about DNA methylation and TF binding, it is important that the threshold for understanding is high enough (which I believe, in its current form, is not the case). Basically, to enable causal relationship between a priori DNA methylation state and reporter output is very difficult to achieve simply because transcription factors are at play.

– Transcription factors can drive cell state transitions (e.g., differentiation, reprogramming, etc.), and are known to "override" DNA methylation state, and also to maintain methylation state. Specificity of regulatory regions is firstly determined by active transcription factors (e.g. GATA in K562). Regions with GATA motifs are found to be active independent of DNA methylation, which is in line with this, since GATA factors play a driving role of K562 identity/transcriptome. If the same library would be transfected into another cell line, with other driving factors, then other enhancers would be predicted to show methylation-independent regulation. The conundrum in this reasoning, is that these driver TFs, when positively regulating reporter gene expression, would result in de-methylation; while the authors report that methylation before and after is unchanged.

– Reasoning further on this point. The fact that enhancer methylation remains "intact" after transfection is not in agreement with current literature, whereby transcription factors that bind to the enhancer, and activate a target gene (here, the reporter), result (directly or indirectly) in loss of methylation. Since such methylation changes are not observed in the presented experiments, there may be at least two possible explanations. Firstly, there could be a technical problem/artefact somewhere, for example in the re-sequencing of the plasmids after transfection, or in the analysis; secondly: the methylation/demethylation could be dependent on chromatin, while the episomal constructs are not chromatinized (they could be chromatinized, but the authors have not tested whether they are). Which of these explanations the authors think is the correct one? This should all be discussed and supported by experiments/analyses. If the authors claim that it is normal that methylation is so stable that changes in transcription do not alter DNA methylation, then they go against existing data, and should support this with additional experiments. For example, induction/over-expression of a transcription factor, or knock-down of a transcription factor, which causes significant transcriptional changes of its target genes, does (unequivocally) alter DNA methylation state of the enhancers/promoters of its target genes (the genes that are directly bound, can be checked with ChIP-seq and motifs) and that are significantly up or down-regulated. In such an experiment, if the endogenous enhancer does changes in DNA methylation, but the episomal construct with a copy of that enhancer does not, then it's clear that methylation changes are not recapitulated in the episomal construct. If, on the other hand, the episomal enhancer also changes, then the conclusion in the current version, that DNA methylation does not change, is incorrect. In conclusion, the entire premise of this work is that TFs are fully dependent on the DNA methylation state; they can either bind unmethylated or methylated DNA, but not alter it. This goes against most literature, where transcription factors "rule", in the sense that over-expression of a particular combination of TFs (e.g., Yamanaka factors), or sometimes one, results in changes of cell state, transcriptome, hence DNA methylation.

– Related to this point again. No strong correlation between methylation and expression was reported before, yet differential methylation does correlate with differential gene expression. For example, an increase in gene expression is correlated with a decrease in methylation of promoter and/or enhancers that underlie this up-regulation. To take this into account, it may be required to measure the reporter assay, and the methylation state, of the reporters during a state transition, such as differentiation, or a treatment (e.g., drug treatment of K562 that causes large transcriptional changes).

– The prediction of motifs in methylation dependent vs. independent enhancers is good, but remains at the prediction level, and is not validated experimentally.

Reviewer #3:

The paper reports a massively parallel reporter assay to adapt STARR-seq to probe methylation dependent regulatory activity. In principle, this assay is innovative and important, as it will be able to address timely and relevant questions. Understanding the functional consequence of DNA methylation variation is a major open challenge, which can be addressed using this assay. Overall, the paper is clear and well written. The technical controls present a largely sufficient. However, I have a few technical comments and would request extended analyses on assay coverage and other aspects of the computational analysis.

1) Improved analysis of sequence coverage. Based on the current description of the text, it is hard to decipher the coverage properties of the method. It would be helpful to clarify the CpG coverage as a function of the number of fragments assayed and the number of replicates taken into account. Also, how uniform is the fragment coverage compared to a complete MspI digest? Are there any "blindspots" of the protocol?

2) Assessment of raw count data and quantification of regulatory activity. Access of plasmid-derived mRNA relative to DNA input is used to define regulatory activity. I would like to see more analysis on the characteristics of these quantitative data. What is the dynamic range of this activity measure, e.g. in their distribution for the 200bp bins that are used for analysis? A related question is additional QC for the linear model that is used to call (MD) regulatory activity. The approach outline in the Materials and methods seems sensible, however I would request additional diagnostics to check that the calls are statistically calibrated. Given that only a small proportion of all windows are regulatory active, the distribution of p-values should be uniform up the ~8% that contain signal. A QQ plot would be reassuring to validate that the filters employed are appropriate.

3) Impact of fragment length. If I interpret Figure 2B correctly, the enrichment of mSTARR-seq enhance activity for enhancer annotations are substantially higher for longer fragments. The authors comment on a relationship of assay sensitivity with fragment length. However, I feel this issue warrants further exploration. In particular, the enrichment analysis is not strictly a measure of sensitivity only but instead a combination of sensitivity and specificity. A more detailed analysis of coverage by fragment size and specificity and sensitivity would seem useful, also to guide the user for designing future studies.

4) Prediction of MF and non-MD regulatory activity. The random forest analysis is interesting. I would request additional controls/clarifications. First, the evaluation approach is somewhat misleading. The results are presented in terms of accuracy at an undefined threshold. A canonical approach would be a precision/recall curve for a continuous range of prediction thresholds. Similarly, a "clean" hold-out assessment, e.g. using a certain proportion of bins that were not used during training, would be much better than using outer bags of the RF. Finally, I am puzzled about the presented results on feature importance. CpG_number, CpP_density and Methylation_level are correlated variables, yet taking out anyone of these predictors does appear to result in a substantial drop in accuracy. It is not clear whether the importance measure accounts for correlations between features and what precisely is shown in Figure 3F.

---

## [Author Response]

[Editors' note: the authors’ plan for revisions was approved and the authors made a formal revised submission.]

Reviewer #1:The authors propose a new experimental approach, mSTARR-Seq, for testing the causal effects of DNA methylation on gene expression at millions of methylation CpG sites. They find that few sites are 'methylation dependent' but those that are MD are predictable from chromatin state measurements. The approach, to my knowledge, is novel, robust and produces important new findings. I do not have substantive concerns. The comments that I provide below are thus not intended to imply a lack of support for this manuscript.I was surprised that the 8% figure in Results paragraph three was not emphasised more. It is important, to my mind, that the reader is told explicitly about this 92%:8% split. Previous statements, such as "our observations indicate a substantial degree of coordination between methylation levels and gene expression" (Banovich et al.) might be interpreted as implying otherwise.

Thanks; we think you are referring here to our estimate of 15% (Results section, paragraph four, which is the percentage of tested regions capable of enhancer activity that are also methylation-dependent), as opposed to the 8% figure in paragraph three (which is the percentage of tested regions with enhancer activity in either the unmethylated or methylated state, in agreement with estimates in previous STARR-seq papers: e.g., Arnold et al., 2013).

We agree that our 15% estimate may be surprising to readers who expect tight coordination between methylation levels and gene expression levels, and we now place greater emphasis on this point in the revised manuscript (Discussion first paragraph). However, we also note that methylation-dependent activity is likely dependent on cell type, cellular environment, and the regions assayed. Further, power to detect MD activity depends on the level of experimental replication; some regions do exhibit greater enhancer activity in one condition relative to the other, but do not reach statistical significance in our study (as now observable in new Figure 2—figure supplement 1). Thus, rather than emphasizing the 15% estimate *per se*, our revisions highlight (i) that MD activity is probably the minority case for CpG sites in the human genome; and (ii) that, where they occur, many MD effects may be weak.

The RF approach predicted MD regulatory element activity with ~79% accuracy, with 26 features being significantly contributing. The MD transcription factor-DNA binding findings (summarised in Figure 4) are complementary to similar findings from others, e.g. Yin et al., 2017 and Table 1.

Thanks; this complementarity is now emphasized in paragraph three of subsection “Determinants of methylation-dependent (MD) regulatory activity” and Figures 5A/B.

"We also show that MD regulatory elements explain site-to-site heterogeneity in the strength of DNA methylation-gene expression correlations measured within a population sample". Can the authors be clearer quantitatively how explanatory this is?

We have revised our Abstract and Results to clarify this point (Abstract section and subsection “mSTARR-seq explains site-to-site heterogeneity in the strength of DNA methylation level-gene expression level correlations in vivo”). CpG sites in MD regulatory elements are 1.6x more common among sites where DNA methylation-gene expression correlations are negative and strong (i.e., FDR < 10% and rho < -0.2) than among other tested sites in the same data set.

Reviewer #2:Lea et al. describe mSTARR-seq: STARR-seq using a library of enhancers, whereby the library is either methylated (in bacteria) or unmethylated. Comparison of enhancer reporter levels between both conditions, allows identifying methylation-dependent enhancers, as well as methylation-independent. What is interesting about this study is that it aims to provide a direct measurement of the causality of DNA methylation of an enhancer/promoter sequence in driving gene expression. However, the interpretation of the data is not fully convincing, there is little follow-up, and because of that the study, in its current form, has limited depth. This is a difficult problem, so the attention and approach is timely, but given the confusion and contradictions in the field about DNA methylation and TF binding, it is important that the threshold for understanding is high enough (which I believe, in its current form, is not the case). Basically, to enable causal relationship between a priori DNA methylation state and reporter output is very difficult to achieve simply because transcription factors are at play.

We appreciate the helpful feedback and agreement on the importance of the problem. We also agree that transcription factors play an integral role in mediating the DNA methylation-gene expression relationship, and that understanding their contribution to the causal chain is essential. We have added several new experiments and analyses (detailed below) to address your specific comments.

– Transcription factors can drive cell state transitions (e.g., differentiation, reprogramming, etc.), and are known to "override" DNA methylation state, and also to maintain methylation state. Specificity of regulatory regions is firstly determined by active transcription factors (e.g. GATA in K562). Regions with GATA motifs are found to be active independent of DNA methylation, which is in line with this, since GATA factors play a driving role of K562 identity/transcriptome. If the same library would be transfected into another cell line, with other driving factors, then other enhancers would be predicted to show methylation-independent regulation. The conundrum in this reasoning, is that these driver TFs, when positively regulating reporter gene expression, would result in de-methylation; while the authors report that methylation before and after is unchanged.– Reasoning further on this point. The fact that enhancer methylation remains "intact" after transfection is not in agreement with current literature, whereby transcription factors that bind to the enhancer, and activate a target gene (here, the reporter), result (directly or indirectly) in loss of methylation. Since such methylation changes are not observed in the presented experiments, there may be at least two possible explanations. Firstly, there could be a technical problem/artefact somewhere, for example in the re-sequencing of the plasmids after transfection, or in the analysis; secondly: the methylation/demethylation could be dependent on chromatin, while the episomal constructs are not chromatinized (they could be chromatinized, but the authors have not tested whether they are). Which of these explanations the authors think is the correct one? This should all be discussed and supported by experiments/analyses. If the authors claim that it is normal that methylation is so stable that changes in transcription do not alter DNA methylation, then they go against existing data, and should support this with additional experiments. For example, induction/over-expression of a transcription factor, or knock-down of a transcription factor, which causes significant transcriptional changes of its target genes, does (unequivocally) alter DNA methylation state of the enhancers/promoters of its target genes (the genes that are directly bound, can be checked with ChIP-seq and motifs) and that are significantly up or down-regulated. In such an experiment, if the endogenous enhancer does changes in DNA methylation, but the episomal construct with a copy of that enhancer does not, then it's clear that methylation changes are not recapitulated in the episomal construct. If, on the other hand, the episomal enhancer also changes, then the conclusion in the current version, that DNA methylation does not change, is incorrect. In conclusion, the entire premise of this work is that TFs are fully dependent on the DNA methylation state; they can either bind unmethylated or methylated DNA, but not alter it. This goes against most literature, where transcription factors "rule", in the sense that over-expression of a particular combination of TFs (e.g., Yamanaka factors), or sometimes one, results in changes of cell state, transcriptome, hence DNA methylation.

We agree that TF binding can lead to changes in the DNA methylation state of regulatory elements. However, TF binding can also itself be methylation-dependent (as shown in Yin et al., 2017, Hu et al., 2013, and supported by the data presented here in Figures 4 and 5A/B). Thus, we now clarify that our data are best interpreted as a test of whether differences in DNA methylation are *sufficient* to causally alter gene regulation, in the context of the cellular environment in which the assay is conducted (Discussion section, paragraph two). Because the treatment condition (methylated versus unmethylated) is the only exogenous variable in an mSTARR-seq experiment, all signals of MD activity depend on the original treatment status—even if methylation status changes after TF binding. In this respect, the assay is conservative because rapid demethylation of the methylated treatment (or rapid methylation of the unmethylated treatment) would result in no MD signal: the two treatment conditions, methylated and unmethylated, would effectively become identical (see paragraph five of subsection “Determinants of methylation-dependent (MD) regulatory activity”).

Nevertheless, we very much appreciate your point about TF remodeling of DNA methylation. Our initial report of DNA methylation stability was based on bisulfite sequencing analysis of CpG sites introduced through Gibson assembly of test fragments ligated to Illumina-style adapters, which contain 2 CpGs. This was described in the Materials and methods (subsection “Generation and low-level processing of bisulfite sequencing libraries”) but is now clarified in paragraph two of the Results section and the legend for Figure 1E. Figure 1E therefore shows that *transfection itself* does not compromise our experimental manipulation of DNA methylation. It does not speak to the methylation status of individual fragments with binding sites for specific TFs. However, we agree that changes in DNA methylation post-transfection are expected near the binding sites of TFs linked to demethylating behavior (to our knowledge, this behavior is well-supported for some, but not all TFs: e.g., Suzuki et al., 2017 and references therein).

To test this prediction and address the concerns raised above, we therefore performed additional experiments to generate a locus-specific bisulfite sequencing data set that allowed us to ask whether and where DNA methylation levels change post-transfection (our earlier data in Figure 1E are too low coverage to address this question). Consistent with our previous findings, at the CpG sites introduced by Gibson assembly (that is, at sites that link fragment inserts to the plasmid backbone), the mean methylation level post-transfection is high for the methylated treatment (95.1%) but very low for the unmethylated treatment (1.6%). Thus, we again confirm that transfection itself does not appreciably disrupt our experimental treatment.

However, for human DNA fragments inserted in the mSTARR-seq vector, we observe low levels of both demethylation (of methylated condition fragments) and the addition of methyl groups (to unmethylated condition fragments). These changes result in convergence to endogenous K562 DNA methylation levels post-transfection (now reported in new Figure 4—figure supplement 1 and paragraph five of subsection “Determinants of methylation-dependent (MD) regulatory activity”), suggesting that the sequence of mSTARR-seq query fragments, in combination with the environment of K562 cells, provides information about cell type “appropriate” DNA methylation levels. Further—and directly relevant to the concerns raised above—specific transcription factor binding sites (based on ChIP-seq data in K562s) are overrepresented in these regions (see new Supplementary File 9) and demethylation is most strongly enriched at GATA TF ChIP-seq peaks. Our data are therefore consistent with the argument that some, but not all, TFs are methylation-sensitive and some, but not all, are associated with subsequent demethylation; however, pioneer TFs like the GATA TFs may have both of these characteristics. We describe these results in paragraph six of subsection “Determinants of methylation-dependent (MD) regulatory activity”, and new Figure 4—figure supplement 1 of the revised manuscript, and thank the reviewer for motivating this important experiment and analysis.

– Related to this point again. No strong correlation between methylation and expression was reported before, yet differential methylation does correlate with differential gene expression. For example, an increase in gene expression is correlated with a decrease in methylation of promoter and/or enhancers that underlie this up-regulation. To take this into account, it may be required to measure the reporter assay, and the methylation state, of the reporters during a state transition, such as differentiation, or a treatment (e.g., drug treatment of K562 that causes large transcriptional changes).

We agree that major changes in cell state will produce shifts in gene expression that correlate with shifts in DNA methylation. Our analysis relating mSTARR-seq to DNA methylation-gene expression correlations focused on DNA methylation and gene expression values measured *across individuals* (at the same loci), whereas DNA methylation-gene expression correlations after changes in cell state focus on correlations *across loci* in a single genome. These are different levels of analysis that produce different results. For example, while the mean correlation across individuals in the monocyte data set is near 0, the mean correlation within a single genome is rho = -0.152 +/- 0.018 s.d. (all p<10^-16^; rho is calculated for each of n=1202 human monocyte data sets).

In the revision, we have clarified these key differences (see subsection “mSTARR-seq explains site-to-site heterogeneity in the strength of DNA methylation level-gene expression level correlations in vivo”). In addition, because we agree that DNA methylation-gene expression correlations across loci following major gene expression shifts are of interest, we have added a new experiment to the manuscript. Specifically, we perturbed K562 gene expression via treatment with IFN alpha, and measured subsequent changes in plasmid DNA methylation and reporter gene expression. These results highlight that large scale changes in the expression of mSTARR-seq fragments (i.e., an increase in mSTARR-seq RNA levels relative to input DNA levels, induced by IFN alpha treatment) are associated with changes in methylation status on the same fragments. Thus, while mSTARR-seq is designed to ask “what happens to gene expression when DNA methylation is perturbed/varies?” rather than “what happens to DNA methylation when gene expression is perturbed/varies?”, it remains entirely consistent with previous observations that differential methylation and differential expression are correlated within a genome, when that genome undergoes a major state transition.

– The prediction of motifs in methylation dependent vs. independent enhancers is good, but remains at the prediction level, and is not validated experimentally.

We apologize for our lack of clarity: we do not perform any prediction of TF motifs in our manuscript. We now clarify that our random forests analysis uses experimentally ascertained TF binding sites from K562 ChIP-seq data (subsection “Determinants of methylation-dependent (MD) regulatory activity” and “Annotation of analyzed mSTARR-seq fragments”), while our analyses of TF motifs in MD and non-MD enhancers and in comparison to SELEX data rely on a library of known TF motifs from the HOMER database (paragraph three of subsection “Determinants of methylation-dependent (MD) regulatory activity” and subsection “Transcription factor binding motif enrichment analyses”). We also integrate independent experimental data in two ways: (i) we compare TFs enriched in MD enhancers that are more active in either methylated or unmethylated states to DNMT triple-knockout DNase-seq data (paragraph four of subsection “Determinants of methylation-dependent (MD) regulatory activity” and Figure 4); and (ii) we compare our enrichments for MD and non-MD enhancers to SELEX data on TF binding to methylated versus unmethylated DNA ligands (paragraph three and Figure 5). Both analyses corroborate our findings, providing a key form of experimental validation even though the other data sets were generated elsewhere.

Reviewer #3:1) Improved analysis of sequence coverage. Based on the current description of the text, it is hard to decipher the coverage properties of the method. It would be helpful to clarify the CpG coverage as a function of the number of fragments assayed and the number of replicates taken into account. Also, how uniform is the fragment coverage compared to a complete MspI digest? Are there any "blindspots" of the protocol?

We agree that these summaries are useful. We now show how the number of fragments assayed, number of CpGs analyzed, and coverage per CpG site scales depending on sequencing depth and the number of technical replicates performed (see new Figure 2—figure supplement 2). With respect to possible ‘blindspots’ of our protocol, we report in Figure 1—figure supplement 4 and Results paragraph three that our final dataset includes 57% of fragments expected from a complete *Msp1* digest of the human genome. However, our plasmid input library contains 93% of expected fragments (Results paragraph two), indicating that the majority of fragments that are “missing” in our analysis are not represented because they are not consistently expressed in either methylated or unmethylated conditions. Our data thus suggest minimal true “blindspots,” but do point to many regions with minimal regulatory activity. We have added this information to the legend for Figure 1—figure supplement 4.

2) Assessment of raw count data and quantification of regulatory activity. Access of plasmid-derived mRNA relative to DNA input is used to define regulatory activity. I would like to see more analysis on the characteristics of these quantitative data. What is the dynamic range of this activity measure, e.g. in their distribution for the 200bp bins that are used for analysis?

To show the dynamic range of enhancer activity, and particularly the range we are able to identify as showing significant enhancer activity, we have added two new plots (Figure 2—figure supplement 1). The first figure shows the distribution of log2 fold changes between normalized RNA and normalized DNA counts (where excess RNA relative to DNA implies enhancer activity) for regions we called as enhancers versus regions without significant regulatory activity. The second shows the proportion of regions we called as enhancers as a function of the log_2_-fold change. As we note above (response to Reviewer 1), these plots show that weak enhancers may be undetected at the significance thresholds we employed

A related question is additional QC for the linear model that is used to call (MD) regulatory activity. The approach outline in Materials and methods seems sensible, however I would request additional diagnostics to check that the calls are statistically calibrated. Given that only a small proportion of all windows are regulatory active, the distribution of p-values should be uniform up the ~8% that contain signal. A QQ plot would be reassuring to validate that the filters employed are appropriate.

We agree that the distribution of p-values should be uniform for the set of true negatives (true non-MD enhancers). However, although our dataset includes more replicates than typically employed in massively parallel reporter assays, our set of regions without significant activity undoubtedly contains false negatives due to power constraints (especially after correction for multiple hypothesis testing and especially for loci with weak enhancer activity: see new paragraph two of subsection “Validation against low-throughput reporter assays and comparison with endogenous measures of regulatory function”). This is evident in the QQ plot in new Figure 3—figure supplement 1. Importantly, however, permutation analyses show that true negatives produce the expected uniform distribution of p-values when using the same modeling approach (Figure 3—figure supplement 1). This result indicates that our statistical tests are well-calibrated, even if we miss weakly MD enhancers.

3) Impact of fragment length. If I interpret Figure 2B correctly, the enrichment of mSTARR-seq enhance activity for enhancer annotations are substantially higher for longer fragments. The authors comment on a relationship of assay sensitivity with fragment length. However, I feel this issue warrants further exploration. In particular, the enrichment analysis is not strictly a measure of sensitivity only but instead a combination of sensitivity and specificity. A more detailed analysis of coverage by fragment size and specificity and sensitivity would seem useful, also to guide the user for designing future studies.

Thanks, this is an excellent point. We now show the trade-off between sensitivity and specificity in revised Figure 2—figure supplement 3, for enhancers found on fragments of different lengths. We note that this analysis is somewhat limited by the assumption that ENCODE enhancer annotations are all true positives and that regions that are not annotated as enhancers are all true negatives. Even with this limitation, however, specificity/sensitivity is still clearly quantitatively related to fragment length: larger fragments are both better at detecting true enhancers and less prone to false positives.

4) Prediction of MF and non-MD regulatory activity. The random forest analysis is interesting. I would request additional controls / clarifications. First, the evaluation approach is somewhat misleading. The results are presented in terms of accuracy at an undefined threshold. A canonical approach would be a precision/recall curve for a continuous range of prediction thresholds.

We apologize for the lack of clarity. We report the proportion of regions that were correctly assigned by the random forests analysis as MD enhancers or non MD enhancers. Each model iteration/classification tree assigns a given region to a given outcome class, and we considered a given region to be assigned to a given outcome overall if the majority (>50%) of trees ‘voted’ for this class (we used 1000 model iterations/classification trees). We now clarify this procedure in subsection “Random forests classification” of the revised Materials and methods and show an ROC curve that explores our results if we require a threshold >50% for class assignment in new Figure 3—figure supplement 2.

Similarly, a "clean" hold-out assessment, e.g. using a certain proportion of bins that were not used during training, would be much better than using outer bags of the RF.

We agree on the importance of using a distinct test set. We set our random forest procedures such that each model iteration used 2/3 of the data for training and a completely independent 1/3 of the data for testing. The set of regions used as the training versus test set are permuted across each of the 1000 model iterations, but within a given run the test data are completely held out during model training, which we now clarify in the revised text (subsection “Random forests classification”).

Finally, I am puzzled about the presented results on feature importance. CpG_number, CpP_density and Methylation_level are correlated variables, yet taking out anyone of these predictors does appear to result in a substantial drop in accuracy. It is not clear whether the importance measure accounts for correlations between features and what precisely is shown in Figure 3F.

Our importance measures do not account for correlations between features. Thus, Figure 3F shows the mean decrease in accuracy when excluding each focal variable in isolation. If the information contained in that focal variable is completely redundant with the information provided by other, correlated variables, the accuracy of the RF will not drop. Although CpG_number, CpG_density and Methylation_level are correlated (Spearman’s rho: CpG_number and Methylation_level, -0.048; CpG_density and Methylation_level, -0.155; CpG_number and CpG_density, 0.91), they are not perfectly collinear. Our results therefore indicate that they contain some independent pieces of information about methylation-dependent activity. We now clarify this point in subsection “Random forests classification”, and note that if we remove CpG_number and CpG_density, our model accuracy drops from 78% to 72%.